# Neuron-Aware Data Selection in Instruction Tuning for Large Language Models

**Xin Chen**[1,2,5,*], **Junchao Wu**[1,*], **Shu Yang**[4], **Runzhe Zhan**[1], **Zeyu Wu**[1], **Min Yang**[2,3,†],
**Shujian Huang**[5], **Lidia S. Chao**[1], **Derek F. Wong**[1,†]

[1]NLP[2]CT Lab, Department of Computer and Information Science, University of Macau,
[2]Artificial Intelligence Research Institute, Shenzhen University of Advanced Technology,
[3]Shenzhen Institutes of Advanced Technology, Chinese Academy of Sciences,
[4]Provable Responsible AI and Data Analytics Lab, KAUST,
[5]National Key Laboratory for Novel Software Technology, Nanjing University
`nlp2ct.{xinchen,junchao,runzhe,zeyu}@gmail.com, min.yang@siat.ac.cn`
`shu.yang@kaust.edu.sa,{derekfw,lidiasc}@um.edu.mo`

## Abstract

Instruction Tuning (IT) has been proven to be an effective approach to unlock the powerful capabilities of large language models (LLMs). Recent studies indicate that excessive IT data can degrade LLMs performance, while carefully selecting a small subset of high-quality IT data can significantly enhance their capabilities. Therefore, identifying the most efficient subset data from the IT dataset to effectively develop either specific or general abilities in LLMs has become a critical challenge. To address this, we propose a novel and efficient framework called NAIT. NAIT evaluates the impact of IT data on LLMs performance by analyzing the similarity of neuron activation patterns between the IT dataset and the target domain capability. Specifically, NAIT captures neuron activation patterns from in-domain datasets of target domain capabilities to construct reusable and transferable neuron activation features. It then evaluates and selects optimal samples based on the similarity between candidate samples and the expected activation features of the target capabilities. Experimental results show that training on the 10% Alpaca-GPT4 IT data subset selected by NAIT consistently outperforms methods that rely on external advanced models or uncertainty-based features across various tasks. Our findings also reveal the transferability of neuron activation features across different capabilities of LLMs. In particular, IT data with more logical reasoning and programmatic features possesses strong general transferability, enabling models to develop stronger capabilities across multiple tasks, while a stable core subset of data is sufficient to consistently activate fundamental model capabilities and universally improve performance across diverse tasks.

## 1 Introduction

IT of LLMs has become a foundational technique for activating LLMs instruction and knowledge capabilities (Ouyang et al., 2022). Previous studies have shown that excessive IT data can degrade LLMs performance, while selecting a small amount of high-quality IT data can significantly improve model performance. For instance, LIMA (Zhou et al., 2023) achieved impressive results using only 1k IT data. However, a major challenge remains as current approaches lack interpretability in identifying "high-quality" data and fail to enhance the specific (one or more) target domain capabilities of LLMs in an open dataset (Wu et al., 2023; Qin et al., 2024; Wang et al., 2024a). Moreover, existing SOTA IT data selection methods like Instruction Mining (Cao et al., 2023), AlpaGasus (Chen et al., 2024) and SelectIT (Liu et al., 2024b) often rely on surface-level features, external models and data for scoring, or the model's uncertainty, which are computationally expensive. This limits the scalability of these methods and their broader application to large-scale data.

---

*Equal contribution.
†Corresponding author.

(a) Neuron Activation Feature Extraction for Target Capability.

(b) Activation Feature-Guided Data Selection.

Figure 1: **Overall framework of NAIT.** First, we capture the neuron activation of the LLM using in-domain data that we want the model to learn. Next, we construct an activation feature through a dimensionality reduction method. Finally, we evaluate the feature alignment score between this activation feature and the model's activation on each candidate dataset to guide data selection.

Inspired by previous works on neuron activations and capabilities of LLMs (Voita et al., 2024; Xu et al., 2024c), we address these concerns in NAIT (**N**euronal **A**ctivation-based efficient **IT** data selection framework), a novel approach designed to evaluate data quality and enhance specific domain capabilities of LLMs by analyzing neuronal activation patterns related to the target domain capabilities of the LLM. The NAIT framework is built on the core hypothesis that when an LLM processes a given sample, the closer the model's resulting neuronal activation pattern aligns with the activation pattern of the target capability, the more effectively that sample enhances the LLM's performance in that specific domain.

NAIT operates in two main stages. First, we extract neuron activation features corresponding to the LLM's target capability. This process begins by providing a small set of in-domain examples that exemplify the specific capability of interest. As the LLM processes these examples, its neuron activation states are recorded to identify reusable activation features linked to the desired capability. These features serve as a representation of the LLM's response and neural activation preferences for the target tasks, forming the foundation for the subsequent data selection phase. Next, we perform capability-driven data selection by identifying and selecting IT data that can further activate and enhance the model's relevant capabilities. This is achieved by comparing the neuron activation features of candidate data within an open and diverse dataset (e.g., Alpaca-GPT4) against the in-domain activation features. By prioritizing data that closely matches the neuron activation patterns of the target capability, NAIT ensures that the selected data is both relevant and effective for improving the model's specialized abilities. In general, NAIT exhibits the following properties: (1) **Efficiency (§4.2 & §5.1 & §5.2 & Appendix H)**: NAIT achieves strong performance with only a small number of in-domain examples and IT data, eliminating the need for external models. It also boasts low computational cost and minimal time requirements. (2) **Robustness (§5.1)**: The framework demonstrates excellent adaptability across different base models, IT datasets, and domain-specific datasets. (3) **Interpretability (§5.3)**: Qualitative analysis shows that NAIT unveils the logical reasoning and programmatic features that possess strong general transferability. Additionally, NAIT identifies a subset of core IT data that remains stable across different task-specific activation features.

Experimental results demonstrate that training LLaMA-2-7b on only 10% of the Alpaca-GPT4 IT dataset selected using NAIT yields an average performance improvement of 3.24% across five tasks compared to training on the entire dataset using IT. Further experiments show that NAIT consistently improves performance across different numbers of in-domain data, base models, diverse IT datasets and selection strategies, highlighting NAIT's strong robustness. The method based on neuron activation features also shows strong interpretability. Further analysis reveals that NAIT tends to select IT data with more logical reasoning and programmatic neuron activation features. Such data can effectively enhance the model's reasoning ability and general ability across various downstream tasks. The key contributions of this paper are as follows:

- Proposing NAIT, the first instruction data selection framework based on neuron activation patterns, introducing a new paradigm for the targeted development of model capabilities.

- Revealing the correlation between neuron activation features and the retention of fundamental model capabilities, while exploring the transferability of these features across different downstream tasks.

- Open-sourcing a cross-task neuron feature library and the Alpaca-NAIT dataset, a high-quality IT dataset curated from the Alpaca-GPT4 dataset using our proposed NAIT.

Table 1: **Comparison of our method and existing methods.** ✔ indicates the presence of a feature, ✘ indicates its absence, and ◯ denotes partial support.

| Method | Feature Source | Externally-Independent | Gradient-Free | Targeted Ability | Cost-Effective | Interpretability |
|---|---|---|---|---|---|---|
| LIMA (Zhou et al., 2023) | Human | ✔ | ✔ | ✘ | ✘ | ✘ |
| Instruction Mining (Cao et al., 2023) | PPL & Reward scores & ... | ✔ | ✔ | ✘ | ✘ | ◯ |
| AlpaGasus (Chen et al., 2024) | ChatGPT Score | ✘ | ✔ | ✘ | ✘ | ✘ |
| SelectIT (Liu et al., 2024b) | Multi-granularity Uncertainty | ✔ | ✔ | ✘ | ✘ | ◯ |
| LESS (Xia et al., 2024) | Gradient-based | ✔ | ✘ | ✔ | ✘ | ◯ |
| NAIT (Ours) | Neuron Activation | ✔ | ✔ | ✔ | ✔ | ✔ |

## 2 RELATED WORK

IT plays a crucial role in bridging general pretraining and task-specific alignment, as summarized in Appendix C. It enables LLMs to follow instructions, activate latent abilities, and significantly enhance downstream task performance.

### 2.1 HIGH-QUALITY IT DATA

Recent studies have highlighted the key role of high-quality IT data in enhancing the alignment of LLMs with human preferences as well as generating accurate, relevant, and safe responses (Wang et al., 2024a; Liu et al., 2024a). Typically, IT data consists of instruction-response pairs, which can be created in two main ways: (1) by reformulating traditional NLP task data into the IT format (e.g., FLAN (Wei et al., 2022) and P3 (Sanh et al., 2022)) or (2) by synthesizing new data using self-instruction based on a small set of manually crafted seed IT data (e.g., Alpaca (Taori et al., 2023)). Subsequent work has further explored methods for synthesizing high-quality IT data. For example, Vicuna (Chiang et al., 2023) enhanced data diversity with large-scale dialogue data, while WizardLM (Xu et al., 2024a) used LLMs to automatically generate IT datasets with controllable complexity. Notably, LIMA (Zhou et al., 2023) demonstrated that LLMs fine-tuned on only 1k carefully curated IT data can achieve comparable performance of models trained with large-scale dataset, highlighting the importance of data quality over quantity. These findings suggest that pretrained LLMs already contain extensive world knowledge and require only a small amount of high-quality instruction data to yield strong performance in the IT phase.

However, defining what constitutes "high-quality" IT data remains challenging. Data quality is influenced by complex, multidimensional factors, and there is still a lack of interpretable selection mechanisms and standardized criteria (Wang et al., 2024a).

### 2.2 EFFICIENT IT DATA SELECTION

Existing IT data selection strategies typically filter subsets based on metrics such as quality, diversity, or importance (Qin et al., 2024). These strategies fall into four main categories based on how these metrics are extracted: (1) *Handcrafted Feature-Based Methods* rely on manually designed features. For example, DQI (Mishra et al., 2020) selects high-quality data by leveraging lexical features, *n*-gram frequency, and relational features, while Xie et al. (2023) optimize data distribution using *n*-gram features. Although these methods more interpretable, they are limited to surface-level characteristics and cannot capture deeper data properties; (2) *Model Feature-Based Methods* extract features such as uncertainty or perplexity from model outputs. For instance, Instruction Mining (Cao et al., 2023) leverages perplexity and reward scores, and SelectIT (Liu et al., 2024b) conducts multi-granularity uncertainty analysis for data selection. However, these methods depend heavily on model outputs, which may introduce bias, and often exhibit "black-box" characteristics that limit interpretability. (3) *LLM-as-Scorer Methods* utilize advanced LLMs, such as ChatGPT, to score data based on complex criteria. For instance, InsTag (Lu et al., 2024) evaluates diversity and complexity in IT data, while AlpaGasus (Chen et al., 2024) scores data quality similarly. These approaches, though effective, are computationally expensive, lack transparency, and are difficult to control or scale due to reliance on closed-source APIs. (4) *Loss and Gradient-based Coreset Sampling Methods* select data based on loss values or gradient information. For example, Chen et al. (2023) select samples to minimize evaluation loss, and Xia et al. (2024) use gradient signals to

identify critical data points. These methods are computationally intensive, especially for large-scale models, and may suffer from approximation errors.

In summary, existing data selection methods often require substantial computational resources, additional model training, or expensive queries, which limit their scalability to large datasets. Furthermore, they may struggle to enhance specific target domain capabilities of LLMs when applied to open datasets.

# 3 OUR NAIT METHOD

## 3.1 PRELIMINARY

Neuron activation analysis provides insights into the knowledge storage and operational mechanisms of LLMs (Voita et al., 2024; Durrani et al., 2020; Xu et al., 2024c). Recent studies show that subsets of neurons are activated by specific tasks, playing a critical role in LLMs' ability to process knowledge and solve various tasks (Yu & Ananiadou, 2024; Wang et al., 2024b; Tang et al., 2024).

**Motivation**   Inspired by these findings, we hypothesize that the effectiveness of instruction data may lie in its ability to activate task-relevant neurons. Specifically, when an LLM processes a given sample, the closer the neuron activation pattern aligns with the activation characteristics of the target capability, the more effective the sample is in improving the LLM's performance in that domain.

**Comparison**   Compared to existing methods, the proposed NAIT approach offers significant advantages, as shown in Table 1. Specifically, NAIT directly leverages neuron activation patterns within LLMs, eliminating the need for external models or complex proxy features and greatly improving data selection efficiency. Furthermore, due to the strong correlation between neuron activation and the internal knowledge and tasks of LLMs, NAIT can effectively improve both the model's specialized and general domain capabilities. By capturing the internal neuron activation states during LLM operation, NAIT also provides robust interpretability for data selection. Overall, NAIT outperforms existing methods in efficiency, scalability, and interpretability, offering a more effective solution for IT data selection.

## 3.2 OUR DATA SELECTION FRAMEWORK

NAIT identifies IT data that effectively enhances specific domain capabilities $\mathcal{C}$ in the LLM $\mathcal{M}$ via two main modules: (a) Neuron Activation Feature Extraction and (b) Activation Feature-Guided Data Selection, as illustrated in Figure 1. The proposed framework is detailed below (the detailed algorithm is provided in Appendix F):

### 3.2.1 MODULE (A). NEURON ACTIVATION FEATURE EXTRACTION FOR TARGET CAPABILITIES

To establish the neuron activation features associated with target capabilities $\mathcal{C}$, we implement a two-stage features extraction process:

- **In-domain Dataset and Activation Capture**
  We aim to analyze the intrinsic representations of capabilities $\mathcal{C}$ by constructing a representative dataset $\mathcal{P} = \{P_i\}$. Given an in-domain sample $P_i = (t_1, \ldots, t_K)$ from $\mathcal{P}$, we record the activations in model $\mathcal{M}$ across the decoder layers $\mathcal{L}$. For a specific layer $l$ and token $t_k$, the activation vector is defined as:
  $$\mathcal{A}(t_k) = [a_j^{(k)}]_{j=1}^J \tag{1}$$
  where $J$ denotes the number of neurons in the layer. Therefore, its relative change $\Delta\mathcal{A}(t_k)$ with respect to the beginning token $b$ can defined as:
  $$\Delta\mathcal{A}_i^{(l)} = \mathcal{A}^{(l)}(t_K) - \mathcal{A}^{(l)}(t_1), \tag{2}$$
  where $\Delta\mathcal{A}(t_k) \in \mathbb{R}^J$ quantifies the dynamic activation shifts for each neuron throughout the generation process. To summarize the activations for the entire sequence, we compute the mean activation across all $K$ tokens.

- **Neuron Activation Direction Extraction**
  To obtain the characteristic direction vectors $\mathcal{V}$, we apply Principal Component Analysis (PCA) to the difference set $\Delta\mathcal{A}^{(l)}$. For each layer $l$, we extract the first principal component:

  $$\mathbf{v}_l = \text{PCA}(\Delta\mathcal{A}^{(l)}). \tag{3}$$

  To ensure the direction $\mathbf{v}_l$ aligns with the capability's activation trend, we compute the mean difference $\mu_{\text{diff}} = \frac{1}{|\mathcal{P}|}\sum(\mathcal{A}^{(l)}(t_K) - \mathcal{A}^{(l)}(t_1))$. We calibrate the direction as follows:

  $$\mathbf{v}_l \leftarrow \begin{cases} -\mathbf{v}_l & \text{if } \mu_{\text{diff}} \cdot \mathbf{v}_l < 0 \\ \mathbf{v}_l & \text{otherwise} \end{cases} \tag{4}$$

  The final set of direction vectors is $\mathcal{V} = \{\mathbf{v}_l\}_{l=1}^{L}$.

### 3.2.2 MODULE (B). ACTIVATION FEATURE-GUIDED DATA SELECTION

Given the instruction dataset $\mathcal{D}_{\text{ins}}$, we perform data selection based on the alignment with the extracted directions $\mathcal{V}$.

- **Activation Feature-guided Data Scoring**
  For each sample $y \in \mathcal{D}_{\text{ins}}$, we calculate a score $s_y$ by projecting the sample's activation onto the target direction $\mathbf{v}_l$ and summing across all layers $l = 1, \ldots, L$:

  $$s_y = \sum_{l=1}^{L}(\mathcal{A}^{(l)} \cdot \mathbf{v}_l) \tag{5}$$

  This score $s_y$ measures how strongly the sample $y$ activates the specific neurons associated with the target capability.

- **Data Ranking & Selection**
  Finally, we select the subset $\mathcal{D}_{\text{selected}}$ containing the top-$k$ samples with the highest scores:

  $$\mathcal{D}_{\text{selected}} = \text{top-}k(\mathcal{S}) \tag{6}$$

## 4 EXPERIMENTS

### 4.1 SETUPS

**Benchmark**  We evaluate NAIT across five representative capability domains using widely adopted benchmarks: factual knowledge with MMLU (Hendrycks et al., 2021) and MMLU-Pro (Wang et al., 2024c), mathematical reasoning with GSM (Cobbe et al., 2021) and SVAMP (Patel et al., 2021), general reasoning with BBH (Suzgun et al., 2023), multilingual understanding with Ty-DiQA (Clark et al., 2020) and XQuAD (Artetxe et al., 2019), and coding ability with BigCodeBench (Zhuo et al., 2024), HumanEval (H-Eval) (Chen et al., 2021) and MBPP (Austin et al., 2021). We use in-domain splits as capability references and conduct evaluation on the corresponding test sets. Benchmark setups are provided in Appendix D.

**Baselines**  We compare NAIT against a diverse set of representative IT data construction and selection approaches: Alpaca-GPT4 (Peng et al., 2023), a widely used self-instruct dataset synthesized by GPT-4; LIMA (Zhou et al., 2023), which shows that a small amount of carefully curated high-quality IT data can be highly effective; AlpaGasus (Chen et al., 2024), which leverages ChatGPT to score and filter data; Q2Q (Li et al., 2024), which evaluates data quality via loss signals from a precursor model; and SelectIT (Liu et al., 2024b), which selects high-quality data by exploiting uncertainty estimates from base LLMs. Detailed descriptions are provided in Appendix D. Additionally, we compare our method with the targeted ability activation approaches, including embedding-based methods, representation-based methods(Zhang et al., 2018; Hanawa et al., 2021), and LESS (Xia et al., 2024), as detailed in Appendix F.

**Implementation Details**  In this study, we adopt LLaMA-2-7b as the foundational model for fine-tuning. The detailed fine-tuning and text generation settings are provided in Appendix D.

Table 2: **Performance comparison of baslines using 10% of the IT data.** This table shows the results of various tasks across multiple baselines. NAIT (e.g., NAIT (MMLU)) refers to the process where in-domain dataset to guide the IT data selection. Random refers to a randomly sampled 10% subset of the IT data. The **Bold** and Underline represent the best and second performance respectively in each column. Δ *(↑)* indicates the performance improvement relative to the ID 01.

| System ID↓ | Method↓ | Factual Knowledge | | Mathematical Reasoning | | Coding Ability | | Multilingual Understanding | | General Reasoning | | |
| | Test→ | MMLU | MMLU-Pro | GSM | SVAMP | H-Eval | MBPP | TydiQA | XQuAD | BBH | AVG | Δ (↑) |
| --- | --- | --- | --- | --- | --- | --- | --- | --- | --- | --- | --- | --- |
| | *Full Fine-tuning* | | | | | | | | | | | |
| 01 | Alpaca-GPT4 (Peng et al., 2023) | 46.87 | 21.89 | 14.63 | 39.00 | 27.87 | **51.58** | 39.48 | 42.99 | 39.94 | 36.03 | - |
| 02 | LIMA (Zhou et al., 2023) | 45.20 | 23.04 | 15.76 | 37.67 | 27.75 | 46.56 | 44.92 | 44.72 | 39.91 | 36.17 | +0.39% |
| 03 | 01 + AlpaGasus (Chen et al., 2024) | 43.21 | 21.96 | 13.34 | 36.67 | 23.94 | 46.08 | 44.70 | 46.84 | 39.91 | 35.18 | -2.34% |
| 04 | 01 + Q2Q (Li et al., 2024) | 46.73 | 21.50 | 14.50 | 35.00 | 25.19 | 44.97 | 44.41 | 48.44 | 40.34 | 35.68 | -0.98% |
| 05 | 01 + SelectIT (Liu et al., 2024b) | **47.90** | 22.86 | 15.40 | 41.11 | 27.92 | 49.47 | 43.91 | 45.56 | 40.33 | 37.16 | +3.15% |
| 06 | 01 + Random Baseline | 47.14 | 21.43 | 14.13 | 35.67 | 25.55 | 47.35 | 44.16 | 46.56 | 39.21 | 35.69 | -0.94% |
| | *Our Proposed Method (Individual Capability Features)* | | | | | | | | | | | |
| 07 | 01 + NAIT (MMLU) | 47.81 | 23.61 | 15.68 | 39.67 | 25.23 | 47.47 | 47.16 | **49.47** | 38.52 | 37.18 | +3.20% |
| 08 | 01 + NAIT (GSM) | 46.45 | **24.50** | 16.00 | **41.33** | 27.84 | 48.41 | 46.54 | 47.97 | 40.28 | **37.70** | **+4.65%** |
| 09 | 01 + NAIT (CodeX) | 47.51 | 23.46 | 15.53 | 38.67 | **28.49** | 49.74 | 44.19 | 46.27 | 39.72 | 37.06 | +2.88% |
| 10 | 01 + NAIT (TydiQA) | 46.17 | 22.82 | 13.80 | 37.67 | 25.02 | 47.88 | **47.78** | 49.23 | 40.00 | 36.71 | +1.89% |
| 11 | 01 + NAIT (BBH) | 47.78 | 23.36 | 13.34 | 36.67 | 25.15 | 47.08 | 45.93 | 48.46 | **40.46** | 36.47 | +1.23% |
| | *Our Proposed Method (All Capability Features)* | | | | | | | | | | | |
| 12 | 01 + NAIT (7 - 11) | 46.83 | 23.29 | **16.53** | 39.67 | 26.44 | 47.62 | 46.09 | 48.27 | 40.02 | 37.20 | +3.24% |

## 4.2 MAIN RESULTS

**Overall Performance** The NAIT method effectively enhances model's overall performance by leveraging neuron activation features across multiple domain capabilities. As shown in Table 2, System 01 represents the baseline performance of the Alpaca-GPT4, while System 12 demonstrates the proposed NAIT method surpassing Alpaca-GPT4 under full fine-tuning setting. Notably, it achieves up to a **4.65%** improvement using mathematical reasoning features (System 08) and a **3.24%** improvement when combining all capability features (System 12). These results significantly surpass existing IT data selection methods, such as AlpaGasus (System 03), Q2Q (System 04), and SelectIT (System 05), while utilizing only 10% of the original data. We also observe that activating specific domain capabilities influences downstream tasks differently. For instance, utilizing only the multilingual capability (System 10) leads to decreased performance on mathematical and coding tasks compared to the baseline. In contrast, when all in-domain capabilities are activated (System 12), the model's mathematical performance notably improves from 14.64 to 16.53, demonstrating the robustness of NAIT.

**NAIT with Distinct Capability Features** The experimental results from System 07 to 11 demonstrate that the NAIT method can achieve more consistent and stable performance improvements when targeting individual-specific abilities. Notably, this advantage is particularly pronounced in complex tasks such as mathematical reasoning, multilingual understanding, and coding ability, with improvements of 7.53% and 6.29%, and 5.33 % over the random baseline, respectively. Further analysis reveals that the contributions of neuron activation features vary across different task domains and exhibit a certain degree of cross-domain transferability. For instance, neuron activation features extracted based on the MMLU in-domain dataset can significantly enhance the model's performance on the multilingual understanding task, while those extracted from the CodeX in-domain dataset can likewise effectively improve the model's performance on the GSM task. It is worth mentioning that the performance of the NAIT method on the GSM dataset reaches as high as 4.65%, even surpassing the model under multi-capability activation (System 12). Additionally, we compare our method with targeted ability activation approaches, including embedding-based methods, representation-based methods (Zhang et al., 2018; Hanawa et al., 2021), and LESS (Xia et al., 2024), as detailed in Appendix E. Overall, NAIT consistently achieves the superior overall average performance across all targeted settings. While gradient-based methods like LESS can outperform in the specific target domain to some extent, this specialization often comes at the cost of generalization, leading to performance degradation in non-target tasks. In contrast, NAIT demonstrates robust transferability, maintaining high proficiency across multilingual and reasoning tasks regardless of the target feature.

# 5 ANALYSIS

This section aims to analyze further and address the following research questions: (1) How do the number of in-domain datasets, the proportion of IT datasets, model size, and IT data methods affect NAIT? (see § 5.1) (2) What are the advantages of NAIT in cost efficiency? (see § 5.2) (3) How interpretable in NAIT? (see § 5.3) (4) How does NAIT perform in target domain? (§ Appendix H)

## 5.1 ABLATION STUDY

**Proportion of Selected IT Dataset** Although NAIT demonstrates excellent performance in data evaluation and ranking, selecting the optimal data proportion from IT datasets with a large amount of redundant information remains challenging. To further investigate this, we adjusted the proportion of the selected IT dataset from 10% to 100% and analyzed the performance of NAIT under different IT data proportions. As shown in Figure 2, the top 30% of IT samples selected by NAIT exhibited the best performance. Notably, as the proportion of training data continued to increase, the model's performance showed an overall downward trend, reaching its lowest point when the entire dataset (100%) was used. These results indicate that excessive redundant data may undermine the model's generalization, further emphasizing the need for data selection to maximize model performance.

**Number of In-domain Data** The number of in-domain data is a critical parameter in our method, as it directly influences the effectiveness of extracting neuron activation features for specific domain capabilities. As shown in Figure 3, we fine-tune the model using IT data selected based on the activation patterns of each in-domain data group and evaluate its performance on corresponding tasks. The results indicate that NAIT outperforms the random selection baseline across all tasks in most cases. Even when the data size is small (e.g., 16, 64, or 256 samples), most cases exceed the random baseline. This suggests that even with limited data, the extracted features effectively capture the required capabilities. As the number of in-domain data increases, performance improvements gradually level off, with most tasks reaching their maximum performance at the largest data size. However, for GSM and TydiQA, peak performance is achieved at 18.35 and 47.57, respectively, when the data size is 4096. This highlights that the quality of in-domain data also plays an important role in enabling NAIT to capture task-specific features.

Figure 2: **Performance at different proportions of the IT dataset.** Task performance in NAIT (the in-domain dataset), e.g., MMLU in NAIT, refers to using in-domain data to guide IT data selection and to assess task outcomes. The comprehensive results correspond to System 12.

**Intensity of Neuron Activation** We evaluate the effectiveness of NAIT by comparing three NAIT settings: **Random** (10% of samples randomly selected), **High** (top 10% with highest neuron activation feature alignment scores), and **Low** (10% with lowest neuron activation feature alignment scores). As shown in Table 3. Specifically, NAIT with highest neuron activation feature alignment scores outperforms the Random baseline by 3.35%, indicating that these samples effectively enhance the models' capabilities. In contrast, NAIT with lowest neuron activation feature alignment scores reduces performance by 17.54% compared to Random baseline, suggesting that such samples not only fail to help but also negatively impact the LLM's effectiveness.

Table 3: **Performance comparison across datasets at 10% IT dataset under different neuron activation levels.**

| Intensity | MMLU | BBH | H-Eval | TydiQA | H-Eval | Overall AVG | Overall Δ (↑) |
|-----------|------|-----|--------|--------|--------|-----|-----|
| Random | **47.14** | 39.21 | 14.13 | 44.16 | 25.55 | 34.04 | - |
| High | 46.83 | **40.02** | **16.53** | 46.09 | **26.44** | **35.18** | +3.35% |
| Low | 46.30 | 28.15 | 10.61 | 36.12 | 20.15 | 28.27 | -17.54% |

Figure 3: **Comparison of NAIT with random data selection across different in-domain data scales at 30% IT dataset.** Task performance in NAIT (in-domain dataset), such as MMLU in NAIT (MMLU), refers to using the in-domain dataset to guide IT data selection and evaluate task performance. The dashed line indicates the 30% of the IT dataset random selection baseline.

**Various Foundation Models** Our method demonstrates significant performance improvements when fine-tuning the *LLaMA-2-7b* model with the curated Alpaca-NAIT dataset. To evaluate its robustness and generalizability, we extend our study to other foundational models, including (1) *LLaMA-2-13b*, representing models of different scales; and (2) *Mistral-7b* and *LLaMA-3-8b*, representing models with different architectures, and (3) *Qwen-2.5-7b*, representing state-of-the-art models with stronger baselines. As shown in Table 4, NAIT consistently enhances performance across all evaluated models. Specifically, it maintains effectiveness on larger scales (*LLaMA-2-13b*) and achieves substantial gains of +21.92% on *Mistral-7b* and +18.65% on *LLaMA-3-8b*. Notably, even on the highly capable *Qwen-2.5-7b*, NAIT further boosts the average performance by +3.83%, outperforming the random baseline and validating its efficacy regardless of the model's initial capabilities.

Table 4: **Effectiveness of NAIT Across Models.** Alpaca and Random are trained on the full dataset and a random of 10% subset, while NAIT selects a 10% subset.

| Method | MMLU | BBH | GSM | TydiQA | H-Eval | Overall | |
|---|---|---|---|---|---|---|---|
| | | | | | | AVG | Δ (↑) |
| *LLama-2-13b* | | | | | | | |
| Alpaca | 53.90 | 45.00 | 20.85 | 44.13 | 34.84 | 39.74 | - |
| +Random | 52.70 | **47.59** | 22.06 | 44.66 | 36.79 | 40.76 | +2.57% |
| +NAIT | **54.10** | 47.04 | **24.11** | **48.89** | **38.53** | **42.53** | **+7.02%** |
| *Mistral-7b* | | | | | | | |
| Alpaca | 47.00 | 40.19 | 12.89 | 35.40 | 34.42 | 33.98 | - |
| +Random | 48.37 | 46.11 | 17.36 | 34.80 | 41.81 | 37.69 | +10.92% |
| +NAIT | **52.90** | **49.07** | **18.95** | **41.22** | **45.02** | **41.43** | **+21.92%** |
| *Llama-3-8b* | | | | | | | |
| Alpaca | 59.60 | 48.38 | 29.11 | 48.38 | 57.36 | 48.57 | - |
| +Random | 48.37 | 55.93 | 37.23 | **54.65** | 69.63 | 52.10 | +7.27% |
| +NAIT | **60.80** | **60.93** | **43.91** | 47.72 | **74.78** | **57.63** | **+18.65%** |
| *Qwen-2.5-7b* | | | | | | | |
| Alpaca | 73.30 | 65.46 | 83.17 | 49.33 | 90.85 | 72.42 | - |
| +Random | 74.00 | 67.78 | **84.31** | 51.25 | 89.63 | 73.39 | +1.34% |
| +NAIT | **74.21** | **67.87** | 83.70 | **58.14** | **92.07** | **75.20** | **+3.83%** |

**Different Instruction Tuning Datasets** To further verify the effectiveness of our method, we extend our IT selection method to other IT datasets, including Evo-Instruct (Xu et al., 2024a) using WizardLM and Orca-GPT4 (Mukherjee et al., 2023). The experimental results are shown in Figure 4, as the proportion of IT data increases, the model performance rises and then declines. Notably, the efficient IT data subsets selected by NAIT achieve optimal performance at 50% (Orca-GPT4) and 80% (Evo-Instruct), significantly outperforming the use of the full dataset. Additionally, Evo-Instruct and Orca-GPT4 require a higher proportion of data, especially Evo-Instruct, which needs up to 80% to achieve optimal performance, reflecting their higher quality, complexity, and information density compared to Alpaca-GPT4.

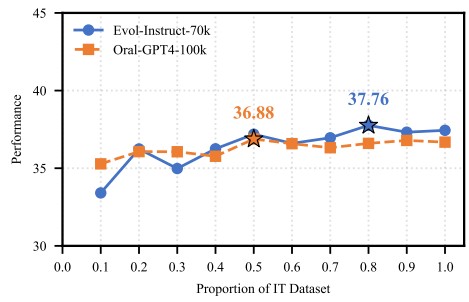

Figure 4: **Comparison of NAIT across different IT dataset.** Task performance in NAIT refers to using the all in-domain dataset to guide IT data selection and evaluate task performance.

**Different Selection Strategies**    We also conducted experiments using context length and perplexity as measures of data complexity, following Liu et al. (2024c), to provide an additional insight into our method's effectiveness, as detailed in Appendix G.

## 5.2    COST EFFICIENCY

We compare inference time and cost between NAIT and existing approaches, including AlpaGasus (GPT-4o), Q2Q (GPT-4o), and SelectIT, on the Alpaca-GPT4 dataset (52k). As shown in Table 5, NAIT achieves up to 19× and 4× cost reductions compared to AlpaGasus and Q2Q, respectively, along with significant improvements in inference speed by 17.75 and 2.2 hours. Compared to SelectIT, NAIT reduces cost by 94.3% and achieves a 17.58× speedup. These results highlight the feasibility and effectiveness of NAIT in real-world applications.

Table 5: **Efficiency Comparison of Different Methods on NVIDIA A800 80GB with Batch Size set to 8**. API costs follow the official OpenAI pricing ($2.00/million input tokens, $8/million output tokens for GPT-4.1); GPU costs are estimated based on the Google Cloud pricing ($1.15 per GPU hour for NVIDIA A800 80GB).

| Method | Externally-Independent | Time | Cost |
|---|---|---|---|
| AlpaGasus(Chen et al., 2024) | ✗ | 19.07h | $178.02 |
| Q2Q(Li et al., 2024) | ✗ | 3.52h | $4.05 |
| LESS(Liu et al., 2024b) | ✓ | 9.86h | $11.33 |
| SelectIT(Xia et al., 2024) | ✓ | 23.20h | $26.68 |
| NAIT | ✓ | 1.32h | $1.52 |

## 5.3    INTERPRETABILITY ANALYSIS

**Transferability of Neural Activation Feature**    From the results in Table 2, we observe that the neural activation features corresponding to different capabilities exhibit remarkable variation in their transferability. For instance, the neural activation feature extracted based on GSM significantly enhances the overall average performance of LLMs, and it also yields positive effects on BBH and CodeX tasks. To further investigate the relationships between the transferability of neural activation features, we introduce **Transferability** to evaluate each capability during the transfer process. Transferability of capability $i$ reflects to what extent $i$ can enhance other capabilities when activated. It is defined as

$$\text{Transferability}_i = \text{Acc}(i, j) - \text{Acc}(j, j) \tag{7}$$

where $\text{Acc}(i, j)$ is the performance on task $j$ activated by feature $i$, and $\text{Acc}(j, j)$ is task $j$'s baseline performance with its own activation feature.

Figure 5: **Transferability of neural activation features across capabilities.** Each column represents the capability to which the neural activation feature is applied. Icons with an outline border indicate the capability's performance on its own task, which serves as the baseline reference.

As shown in Figure 5, the transferability across capabilities can be directly observed. On the one hand, activation features of single capabilities such as GSM and CodeX exhibit evident positive transfer in cross-task settings, indicating that logical reasoning and programmatic features possess strong general power. In contrast, the activation features of TydiQA show weaker or even negative transfer, and their performance within the task itself is relatively limited, reflecting both a dependence on language-specific patterns and shortcomings in cross-domain adaptability. On the other hand, when multiple capabilities are aggregated into a comprehensive activation direction, the model achieves optimal overall transferability. This finding demonstrates the existence of a universal core feature whose neural activation patterns remain stable across tasks, thereby providing a solid foundation for the model's integrated capabilities.

**Direction of Activation Features**    To further investigate the interpretability of NAIT, we apply T-SNE to project the neuron activation features into representation space (see Figure 6). The figure shows that the feature's directions corresponding to different targeted capabilities form relatively separable clusters in the representation space, indicating their consistency with the task-specific reasoning mechanisms or knowledge domains. Notably, the comprehensive, as an ag-

gregated representation across multiple tasks, reveals the **general core capabilities.** This indicates that there exist general data or features capable of simultaneously activating multiple distinct abilities, thereby supporting cross-task knowledge transfer and capability integration.

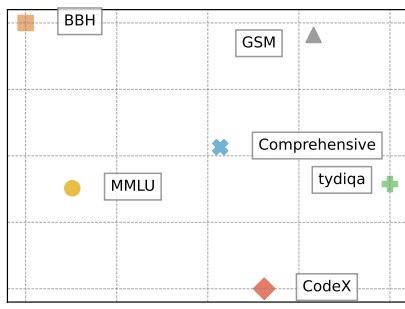

**Selected IT Data Distribution**   To further analyze the distribution of IT subsets selected by different activation features, we performed both qualitative and quantitative analyses (see Appendix I). Figure 7 shows the distribution of the 10% IT dataset selected by NAIT. The results reveal that 58.87% of the subsets overlap across different capabilities, indicating that NAIT consistently identifies a stable set of **general core data** applicable across tasks. However, GSM8K exhibits the highest demand for task-specific data, with 1,034 unique samples, while other tasks like MMLU and CodeX rely more heavily on shared core data. This demonstrates NAIT's ability to balance general-purpose data selection with the retention of task-specific samples, ensuring robust performance across diverse benchmarks. Qualitatively, unlike embedding-based methods that focus on surface-level similarity, NAIT prioritizes samples requiring complex task-following and logical reasoning. While all methods capture the fundamental formats associated with factual knowledge, they consistently select the corresponding factual multiple-choice questions.

Figure 6: **Comparison of the direction of activation feature extracted by NAIT across different capabilities.**

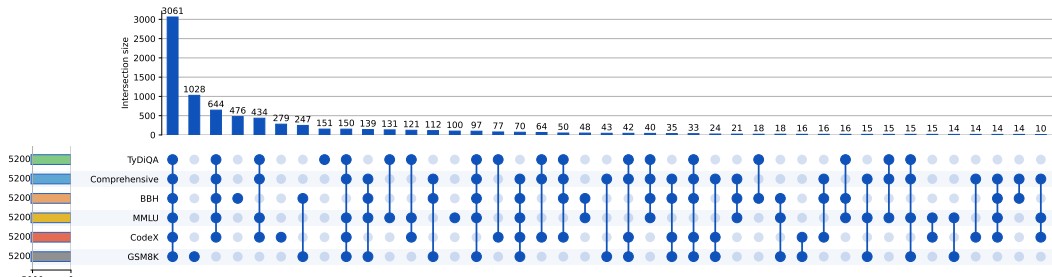

Figure 7: **Distribution of 10% IT dataset selected by NAIT.** The upper bar chart quantifies the intersection sizes among different IT data subsets, while the connected dots below identify the specific capability combinations corresponding to each intersection.

## 6   CONCLUSION

In this paper, we propose NAIT, an efficient framework for selecting high-quality IT data by leveraging neural activation patterns. NAIT identifies the optimal data subset by evaluating the alignment between candidate samples and neural activation features associated with target capabilities. Experimental results demonstrate that models fine-tuned on selected data consistently outperform baseline methods. Ablation studies further verify that only a small number of samples are sufficient to accurately extract activation features that capture model capabilities. In addition, NAIT exhibits strong robustness and reveals significant cross-task transferability of neural activation features. Further analysis shows that data with logical reasoning and programmatic characteristics tend to activate stronger general capabilities in the model, and that a stable core subset of data exists that can universally enhance performance across a variety of tasks.

## ETHICS STATEMENT

This work adheres to the ICLR Code of Ethics.[1] Our research focuses on improving the efficiency and interpretability of IT for LLMs through neurally informed data selection. We do not introduce new data collection involving human participants, nor do we use private or personally identifiable data. All datasets employed (e.g., Alpaca-GPT4, GSM8K, MMLU, BBH, TyDiQA, CodeX) are publicly available and widely used in the research community. We carefully respect data licenses and ensure compliance with privacy and intellectual property standards. Our method aims to reduce the computational and environmental cost of developing LLMs by selecting smaller subsets of high-quality data, which aligns with the principle of minimizing harm. However, as with other LLM research, there is the potential risk of misuse in generating biased or harmful outputs. To mitigate this, we explicitly analyze fairness and generalization across different task domains and report transferability results that highlight both strengths and limits of our approach. We disclose all relevant details transparently to foster responsible use of our method.

## REPRODUCIBILITY STATEMENT

We have taken several steps to ensure reproducibility of our results. Firstly, we describe the construction process of our neuron activation feature framework in § 3 with explicit mathematical definitions. Additionally, detailed descriptions of the model architectures, datasets, hyperparameters, and training setups are provided in § 4.1 and § Appendix D. Further experimental details, including ablation studies and comparisons with baselines, are presented in § 5 and § Appendices G to H. All in-domain datasets and IT datasets used in our experiments (e.g., Alpaca-GPT4, Evo-Instruct, Orca-GPT4) are publicly available, and dataset statistics are summarized in § Appendix D. To facilitate reproducibility, we commit to open-sourcing the implementation of NAIT, the cross-task neuron feature library, and the Alpaca-NAIT dataset upon acceptance. These resources, along with clear documentation, will allow researchers to fully reproduce and extend our experiments.

## ACKNOWLEDGMENTS

This work was supported by National Key Research and Development Program of China (2024YFF0908200), National Natural Science Foundation of China (Grant No. 62376262), the Natural Science Foundation of Guangdong Province of China (2024A1515030166, 2025B1515020032). This work was also supported in part by the Science and Technology Development Fund of Macau SAR (Grant Nos. FDCT/0007/2024/AKP, EF2024-00185-FST), the UM and UMDF (Grant Nos. MYRG-GRG2024-00165-FST-UMDF, MYRG-GRG2025-00236-FST), the Tencent AI Lab Rhino-Bird Research Program (Grant No. EF2023-00151-FST), the Stanley Ho Medical Development Foundation (Grant No. SHMDF-AI/2026/001), and the National Natural Science Foundation of China (Grant No. 62266013).

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

## A  LIMITATION

**Model Scale**  Due to computational limitations, our analysis is restricted to models with fewer than 20B parameters. Future studies could evaluate this method on larger models to provide insights into its scalability and broader implications for optimizing LLMs.

## B  THE USE OF LARGE LANGUAGE MODELS (LLMS)

**Polishing the Writing with a Large Language Model**  In preparing this paper, we used a large language model to refine the writing. Typical applications included: (1) enhancing grammar and style; (2) ensuring consistency of terminology; and (3) improving the quality of translations.

## C  COMPARATIVE ANALYSIS BEFORE AND AFTER IT

Table 6: Comparison of model functions before and after instruction tuning.

| Function | Before Instruction Tuning | After Instruction Tuning |
|---|---|---|
| Enhancing instruction-following ability | Limited to text completion | Capable of following instructions |
| Activating latent abilities | Relies on knowledge from pertaining | Activates abilities not evident in pretraining, such as logical reasoning or task decomposition |
| Enhancing downstream task performance | General knowledge capabilities | Improved downstream task capabilities |

The following table 6 compares the main functional differences of models before and after IT. As shown, IT significantly improves instruction-following abilities, activates latent skills such as reasoning and task decomposition, and leads to notable gains in downstream task performance.

**Benchmark Settings**  We evaluated our approach on various downstream tasks using their data as in-domain data (detailed dataset descriptions are in Appendix D):

- *Factual Knowledge*: We evaluate the Massive Multitask Language Understanding (MMLU) dataset (Hendrycks et al., 2021), splitting it into in-domain and test sets, and report 5-shot results. To assess the generalization capability of our model, we additionally conduct experiments on MMLU-Pro (Wang et al., 2024c) as an OOD benchmark.
- *Math Capability*: We use the Grade School Math (GSM) dataset (Cobbe et al., 2021), divided into in-domain and test sets, and report 5-shot results under the CoT setting. To assess the generalization capability of our model, we additionally conduct experiments on SVAMP (Patel et al., 2021) as an OOD benchmark.

- **Reasoning**: We assess the Big-Bench-Hard (BBH) dataset (Suzgun et al., 2023), with in-domain and test splits, and report 5-shot results under the CoT setting.
- **Multilingual Understanding**: We evaluate the TyDiQA benchmark (Clark et al., 2020), a multilingual QA dataset, using in-domain and test splits, and report 5-shot results with the gold-passage setup. To assess the generalization capability of our model, we additionally conduct experiments on XQuAD (Artetxe et al., 2019) as an OOD benchmark.
- **Coding Capability**: We activate the coding ability using BigCodeBench (Zhuo et al., 2024) and evaluate in HumanEval (Chen et al., 2021), reporting pass@10 results with a sampling temperature of `0.8`. To assess the generalization capability of our model, we additionally conduct experiments on MBPP (Austin et al., 2021) as an OOD benchmark.

## D  DETAILS OF TRAINING AND EVALUATION SETUP

**Statistics of the in-domain, IT, and Test dataset**  Table 7 summarizes the statistics of the In-domain, IT, and Test datasets.

Table 7: **Statistics of the Number of in-domain, IT, and Test datasets Used in Experiment.**

| Targeted Ability→ | Factual Knowledge | | Mathematical Reasoning | | Coding Ability | | Multilingual Understanding | | | General Reasoning |
|---|---|---|---|---|---|---|---|---|---|---|
| **DownStream Task** | **MMLU** | **MMLU-Pro** | **GSM** | **SVAMP** | **HumanEval** | **BigCodeBench** | **MBPP** | **TydiQA** | **XQuAD** | **BBH** |
| **In-domain Dataset** | 1,531 | - | 7,473 | - | - | 1,140 | - | 49,400 | - | 4,556 |
| **Alpaca-GPT4 IT Dataset** | | | | | 52,002 | | | | | |
| **Evol-Instruct IT Dataset** | | | | | 70,000 | | | | | |
| **Oral-GPT4 IT Dataset** | | | | | 100,000 | | | | | |
| **Test Dataset** | 14042 | 2800 | 1319 | 900 | 164 | - | 974 | 900 | 1,200 | 1080 |
| **Test few-Shot** | | 5 | | 8 | | 0 | | | 1 | 1 |

**Baselines Settings**  Our comparison baselines cover a variety of advanced methods for evaluating and improving IT datasets:

- **Alpaca-GPT4** (Peng et al., 2023): A widely utilized IT dataset that leverages a self-instruct methodology, enabling the autonomous generation of instructions through the advanced capabilities of GPT-4.
- **LIMA** (Zhou et al., 2023): It primarily comprises 1,000 meticulously crafted, high-quality IT data points designed to enhance the alignment capabilities of LLMs.
- **AlpaGasus** (Chen et al., 2024): This approach employs the advanced capabilities of ChatGPT to evaluate and selectively curate data from the original Alpaca dataset.
- **Q2Q** (Li et al., 2024): It functions by training a precursor model and assessing the quality of the instructional data based on two distinct loss values derived from this model.
- **SelectIT** (Liu et al., 2024b): A novel method that leverages the intrinsic uncertainty of LLMs to select high-quality IT data without requiring external resources.

**Fine-tuning Parameters**  The model is fine-tuned over 3 epochs with a batch size of `128` to ensure efficient learning while avoiding overfitting. The optimization process employs the Adam optimizer with hyperparameters set to $\beta_1 = 0.9$ and $\beta_2 = 0.999$, which are standard values for stable and effective training. The learning rate follows a cosine decay schedule, starting at `2e-5` and decreasing gradually to `0`, a strategy known to improve convergence by reducing oscillations during later training stages. To maximize the model's performance, we utilize an input sequence length of `2048` tokens, as this configuration has been demonstrated to be effective for LLMs in handling long-context tasks. Table 8 provides an overview of the key hyperparameters used during training on `4 A800` GPUs with `1` node. For the main experiments, we train for 3 epochs with a batch size of 128. For ablation studies, we reduce the training to 1 epoch and adjust the batch size (32 for Llama-2-14B, 64 for Llama-3-8B) to accommodate different model sizes.

**Text Generation Settings**  For text generation, we employ greedy search for computational efficiency, and set the temperature to `0.8` and the top-p to `0.95` following the HumanEval set-

Table 8: **Hyperparameter settings for supervised fine-tuning.**

| Parameter Key | Value | |
|---|---|---|
| | **Main Experiment** | **Ablation Experiment** |
| **Learning rate** | 2.0e-5 | |
| **Cutoff length** | 2048 | |
| **LR scheduler** | cosine | |
| **bf16** | True | |
| **Warmup ratio** | 0.03 | |
| **Weight decay** | 0.1 | |
| **Epoch** | 3 | 1 |
| **Batch size** | 128 | 32 (14B) / 64 (8B) |

Table 9: **Performance comparison of baselines of activating targeted ability using 10% of the IT data.** (e.g., +MMLU refers to the process where an in-domain dataset to guide the IT data selection.) The **Bold** represents the best performance respectively in each column.

| Method↓ | Factual Knowledge | | Multilingual Understanding | | General Reasoning | |
|---|---|---|---|---|---|---|
| Test→ | **MMLU** | **MMLU-Pro** | **TydiQA** | **XQuAD** | **BBH** | **AVG** |
| *+MMLU* | | | | | | |
| Embedding | 46.29 | 21.07 | 42.97 | 44.73 | 36.67 | 38.35 |
| Representation-based | 45.57 | 23.0 | 46.78 | 46.85 | 38.15 | 40.07 |
| LESS | **48.12** | **24.50** | 43.42 | 45.51 | 38.15 | 39.94 |
| Nait | 47.81 | 23.61 | **47.16** | **49.47** | **38.52** | **41.31** |
| *+TydiQA* | | | | | | |
| Embedding-based | **47.49** | 22.54 | 42.4 | 46.53 | 38.43 | 39.48 |
| Representation-based | 45.91 | 21.04 | 40.02 | 46.0 | 36.02 | 37.80 |
| LESS | 44.68 | 22.21 | 45.24 | 47.67 | 37.69 | 39.50 |
| Nait | 46.17 | **22.82** | **47.78** | **49.23** | **40.00** | **41.20** |
| *+BBH* | | | | | | |
| Embedding | 47.51 | 22.86 | 43.25 | 46.66 | 36.76 | 39.41 |
| Representation-based | 45.81 | 21.11 | 36.78 | 44.61 | 37.59 | 37.18 |
| LESS | 46.85 | **23.75** | **46.56** | 47.67 | 39.72 | 39.61 |
| Nait | **47.78** | 23.36 | 45.93 | **48.46** | **40.46** | **41.20** |

ting (Chen et al., 2021), to enhance the creativity and diversity of the generated content while maintaining its accuracy and contextual relevance.

# E  NAIT VS. OTHER BASELINE WITH ACTIVATING TARGETED ABILITY

In this section, we elaborate on the experiment setting and analysis between NAIT and other Baseline with activating targeted ability, incuding embedding-based methods, representation method (Zhang et al., 2018; Hanawa et al., 2021) the gradient-based method (LESS (Xie et al., 2023)).

**Experiment Setting**

- *Embedding-based Method*: We utilize the base model (Llama-2-7b-hf) to extract the static input embeddings. Specifically, we compute the mean of the embeddings across all tokens in the sequence to obtain a global vector representation for each data sample.

- *Representation-based Method* (Zhang et al., 2018; Hanawa et al., 2021): We extract the hidden states of all tokens from the final decoder layer and apply mean pooling to derive a deep semantic representation of the input.

- **LESS** (Xie et al., 2023): As a representative gradient-based approach, LESS selects samples by estimating their influence on the target task via low-rank gradient embeddings. We utilize conversation format of Alpaca-GPT4[2] as the candidate pool. Following the standard of our experiment, we first perform a 3 epoch LoRA warmup to obtain feature checkpoints. Then, using BBH, TydiQA, and MMLU provided by LESS as target validation sets for gradient estimation, we compute influence scores and select the top-$k$ samples from Alpaca-GPT4 that are expected to minimize the validation loss on these targets.

**Result Analysis**  While LESS achieves high scores on targeted tasks (e.g., 48.12 on MMLU vs. NAIT's 47.81), it suffers from overfitting to the target validation set. This results in poor generalization, as evidenced by its lower average score across all tasks (39.94) compared to NAIT (41.31). NAIT maintains robust performance across both targeted and non-targeted settings (e.g., TydiQA and BBH), demonstrating superior transferability. Embedding-based and representation methods rely on surface-level semantic similarity. While computationally cheap, they often fail to capture complex task-specific capabilities. Gradient-based methods (e.g., LESS) calculate influence functions involving gradients and Hessian approximations. This approach is computationally prohibitive and memory-intensive for large-scale models, despite its precision on specific targets.

## F  ALGORITHMIC FRAMEWORK

In this section, we present the detailed pseudo-code for our proposed method, as discussed in Section 3.1. Algorithm outlines the complete procedure, including the extraction of neuron activation features and the subsequent activation feature-guided data selection.

## G  THE DETAILS OF DIFFERENT SELECTION STRATEGIES

Table 10 compares selection strategies using 10% of the IT dataset. NAIT achieves the best performance with a +3.00% improvement over the baseline (Alpaca-GPT4), while other strategies like Hard (PPL) and Easy (Length) significantly reduce performance. This highlights NAIT's effectiveness and the importance of careful data selection.

Table 10: **Performance comparison of different selection strategies using 10% of the IT data.** The **Bold** represents the best performance respectively in each column. $\Delta$ *(↑)* indicates the performance improvement relative to the ID 01 baseline.

| Method | MMLU | BBH | GSM | TydiQA | CodeX | Overall | |
|---|---|---|---|---|---|---|---|
| | | | | | | AVG | $\Delta$ *(↑)* |
| Alpaca-GPT4 | **46.87** | 39.94 | 14.63 | 39.48 | 27.87 | 34.16 | - |
| + Hard (PPL) | 45.52 | 22.22 | 8.72 | 27.87 | 20.56 | 24.98 | -26.87% |
| + Easy (PPL) | 46.78 | 38.15 | 15.39 | 33.12 | 28.93 | 32.47 | -5.53% |
| + Hard (Length) | 46.10 | 36.94 | 16.00 | 34.32 | **30.89** | 32.85 | -3.83% |
| + Easy (Length) | 44.15 | 10.28 | 6.37 | 28.07 | 17.00 | 21.17 | -38.03% |
| + NAIT | 46.83 | **40.02** | **16.53** | **46.09** | 26.44 | **35.18** | **+3.00%** |

## H  NAIT FOR CROSS-LINGUAL

As shown in Table 11, we further evaluate the cross-lingual data selection capability of NAIT on the multilingual IT dataset in the context of machine translation. We adopt the powerful ALMA-7b (Xu et al., 2024b) model as the backbone and conduct experiments on four representative translation directions: English-German, German-English, Chinese-English, and English-Chinese. The experimental results demonstrate that the NAIT method outperforms both random selection (Rand.) and using the full dataset (Full) on most evaluation metrics. Notably, NAIT achieves the highest overall

---

[2]https://huggingface.co/datasets/liangxin/Alpaca_GPT4

---

**Algorithm 1** Neuron-aware Instruction-tuning Data Selection

---

**Input:** In-domain data $\mathcal{P}$, Instruction data $\mathcal{D}_{\text{ins}}$, Model $\mathcal{M}$, Activation layers $\mathcal{L}$, Selection budget top-$k$

**Output:** Selected subset $\mathcal{D}_{\text{selected}}$

1: **Stage 1: Targeted Ability Activation Capture**
2: Initialize activation difference set $\Delta\mathcal{A}^{(l)} \leftarrow \emptyset$ for each layer $l \in \mathcal{L}$
3: **for** each in-domain sample $P_i = (t_1, t_2, \ldots, t_K) \in \mathcal{P}$ **do**
4:      **for** each layer $l \in \mathcal{L}$ **do**
5:          Extract activations $\mathcal{A}^{(l)}(t_1)$ and $\mathcal{A}^{(l)}(t_K)$ from $\mathcal{M}$
6:          $\Delta\mathcal{A}_i^{(l)} \leftarrow \mathcal{A}^{(l)}(t_K) - \mathcal{A}^{(l)}(t_1)$      $\triangleright$ Compute activation change from first to last token
7:          $\Delta\mathcal{A}^{(l)} \leftarrow \Delta\mathcal{A}^{(l)} \cup \{\Delta\mathcal{A}_i^{(l)}\}$
8:      **end for**
9: **end for**
10: **Stage 2: Direction Extraction via PCA**
11: Initialize direction vectors $\mathcal{V} \leftarrow \emptyset$
12: **for** each layer $l = 1, \ldots, L$ **do**
13:      $\mathbf{v}_l \leftarrow \text{PCA}(\Delta\mathcal{A}^{(l)})$      $\triangleright$ Get First Principal Component
14:      $\mu_{\text{diff}} \leftarrow \frac{1}{|\mathcal{P}|}\sum_{i=1}^{|\mathcal{P}|}(\mathcal{A}^{(l)}(t_K) - \mathcal{A}^{(l)}(t_1))$
15:      **if** $\mu_{\text{diff}} \cdot \mathbf{v}_l < 0$ **then**      $\triangleright$ Align Direction With Activation Feature
16:          $\mathbf{v}_l \leftarrow -\mathbf{v}_l$
17:      **end if**
18:      $\mathcal{V} \leftarrow \mathcal{V} \cup \{\mathbf{v}_l\}$
19: **end for**
20: **Stage 3: Activation Feature-guided Data Scoring**
21: Initialize scores $\mathcal{S} \leftarrow \emptyset$
22: **for** each sample $y \in \mathcal{D}_{\text{ins}}$ **do**
23:      Extract Activation $\mathcal{A}^{(l)}$ using $\mathcal{M}(y)$
24:      $s_y \leftarrow \sum_{l=1}^{L}(\mathbf{A}^{(l)} \cdot \mathbf{v}_l)$      $\triangleright$ Project Instruct-tuning Data Onto Target Direction
25:      $\mathcal{S} \leftarrow \mathcal{S} \cup \{s_y\}$
26: **end for**
27: **Stage 4: Data Ranking & Selection**
28: $\mathcal{D}_{\text{selected}} \leftarrow \text{top-}k(\mathcal{S})$
29: **return** $\mathcal{D}_{\text{selected}}$

---

Table 11: **Overall results on machine translation with LLMs using the ALMA-7b.** NAIT indicates selecting the top 30% of data by NAIT, Rand. refers to randomly selecting 30% of data, and Full uses the full dataset.

| Method | En⇒De | | De⇒En | | Zh⇒En | | En⇒Zh | | Overall | |
|---|---|---|---|---|---|---|---|---|---|---|
| | COMET | BLEU | COMET | BLEU | COMET | BLEU | COMET | BLEU | COMET | BLEU |
| + Full | 84.54 | 28.24 | **83.13** | **29.06** | 79.80 | 22.89 | **84.15** | 34.58 | 82.91 | 28.69 |
| + Rand. | 82.44 | 28.29 | 81.12 | 27.20 | 78.53 | 21.11 | 82.12 | 33.91 | 81.05 | 27.63 |
| + NAIT | **85.14** | **29.02** | 82.50 | 28.70 | **80.22** | **23.11** | 83.89 | **35.92** | **82.94** | **29.19** |

COMET (82.94) and BLEU (29.19) scores, further highlighting its superior generalization ability and robustness in both target domains and cross-lingual scenarios.

## I   CASE STUDY

To intuitively demonstrate the effectiveness of our method, we provide a qualitative comparison of the training samples selected by Embedding, LESS, and NAIT for different downstream benchmarks in Table 12 As observed in the table, baseline methods often select data based on surface-level textual similarity, which may not align with the complexity of the target task. For instance, in TydiQA, the embedding method selects translation tasks but may not align with the required question answering.

Table 12: Qualitative comparison of examples selected by Embedding, LESS, and NAIT strategies across different benchmarks. The samples here from NAIT are outside of the intersection of general core data.

| **Targeted ability activated by TydiQA** | | |
|---|---|---|
| **Selected by Embedding** | **Selected by LESS** | **Selected by NAIT** |
| **Instruction:** Translate the phrase "Yechh" into Hindi.

**Output:** The phrase "Yechh" is an expression of disgust or dislike and is not a word with a direct translation into Hindi. However, a similar expression of disgust in Hindi would be "घहअहअरू" or "चहइइ" (Chhee) or "चअधहअम" (chidam) or "अरए बअअपअ रए" (arey baap re). | **Instruction:** Construct a mathematical equation for the following problem:
At a party, there are 3 types of food. Each type of food is served three times each. Guests can choose only 1 of each food serving. How many different combinations are there for the guests?
**Output:** number of combinations = 3 x 3 x 3 = 27 | **Instruction:** Create two authentic Thai dishes.
**Output:** Great! Here are the two Thai dishes!
1. Thai Green Curry Chicken
Ingredients:
- 1 1/2 lbs boneless, skinless chicken breasts
- 1 can (13/14-ounce) coconut milk
- 1/4 cup green curry paste
[...] |
| **Targeted ability activated by BBH** | | |
| **Selected by Embedding** | **Selected by LESS** | **Selected by NAIT** |
| **Instruction:** Given a document, find the most frequently used words.
**Input:** John went to the store to buy some candy.
**Output:** Based on the given sentence, the most frequently used words are:
- "to" - which appears 2 times in the sentence.
- "the" - which appears 1 time in the sentence.
- "John" - which appears 1 time in the sentence.
- "went" - which appears 1 time in the sentence.
- "store" - which appears 1 time in the sentence.
- "buy" - which appears 1 time in the sentence.
- "some" - which appears 1 time in the sentence.
- "candy" - which appears 1 time in the sentence. | **Instruction:** Convert the following document into a list of five bullet points.
**Input:** Sports play a key role in the lives of many people. It can teach young people about leadership and teamwork, as well as important values like perseverance and camaraderie. Sporting events bring communities together and also provide an opportunity for people to come and support their favourite teams or players.
**Output:** - Sports play a key role in the lives of many people
- Sports can teach young people about leadership and teamwork, as well as important values like perseverance and camaraderie.
- Sporting events bring communities together.
- Sporting events provide an opportunity for people to come and support their favourite teams or players.
- Sports promote important values and bring people together. | **Instruction:** Design an algorithm to solve the Eight Queens puzzle
**Output:** The Eight Queens puzzle is a classic problem where the goal is to place eight queens on an 8x8 chessboard in such a way that no two queens threaten each other. This means that no two queens can be placed in the same row, column, or diagonal.
One solution to this problem is to use backtracking. Here is an algorithm to solve the Eight Queens puzzle using backtracking:
1. Start with an empty 8x8 chessboard and add the first queen to the first row in the first column.
2. Move to the next row and try placing a queen in each column. Check if the position is safe by verifying if any of the previously placed queens is attacking the current position. If it's a safe position, place a queen in that column.
3.Boolean function is safe(board, row, col):
[...] |
| **Targeted ability activated by MMLU** | | |
| **Selected by Embedding** | **Selected by LESS** | **Selected by NAIT** |
| **Instruction:** Rank the following four choices in order from least likely to most likely.
**Input:** A. Taking a long walk
B. Eating two slices of chocolate cake
C. Writing a persuasive essay
D. Drinking a glass of orange juice
**Output:** It's not possible to rank the given choices in order from least likely to most likely without additional context or information about the person's habits, preferences or circumstances. Could you provide more information so I can better assist you? | **Instruction:** Choose the best answer from the given list of choices.
**Input:** Question: Which of the following is a benefit of cloud computing?
A. Lower cost
B. Reliability
C. Increased scalability
D. All of the above
**Output:** D. All of the above. | **Instruction:** Generate an example of a multiple choice question related to the following sentence.
**Output:** Entomology is the science which deals with:
A) human behavior
B) the study of insects
C) marine life
D) the solar system
Correct answer: B |

For the general reasoning ability, the Embedding and LESS strategies select simple tasks such as word frequency counting or text summarization. In contrast, NAIT identifies an algorithmic problem that requires multi-step logical reasoning and coding synthesis. Similarly, for MMLU, NAIT selects a knowledge-style question similar to other methods that aligns well with the knowledge-intensive of the benchmark.

