# OpenReview forum: "Neuron-Aware Data Selection in Instruction Tuning for Large Language Models"
_ICLR.cc/2026/Conference — ICLR 2026 Poster_

### Official Review · Reviewer_76SK · 2025-10-28

**Soundness:** 2
**Presentation:** 3
**Contribution:** 2
**Rating:** 4
**Confidence:** 4

**Summary:**

The paper proposes NAIT, a novel and efficient framework for selecting high-quality instruction tuning (IT) data based on the similarity of neuron activation patterns between candidate data and a target capability's in-domain reference set.

**Strengths:**

1. Clarity and Presentation: The paper is well-written and easy to follow.

2. Relevance of Core Idea: The paper successfully addresses the critical insight that the quality of instruction data is more important than the quantity. The proposed method offers a potential, scalable approach for constructing high-quality instruction datasets.

**Weaknesses:**

1. The central hypothesis, that the effectiveness of instruction data lies in its ability to activate task-relevant neurons, is merely an assumption and is not widely recognized or proven.

2. The use of a small, in-domain reference dataset to extract the neuron activation features (for the target capability) may introduce significant bias into the feature extraction process, potentially limiting the generality of the resulting selection criterion.

3. The primary base model used for experimentation is LLaMA and Mistral. Compared to other similarly-sized open-source LLMs (Qwen2.5, Qwen3) available today, LLaMA and Mistral series are generally considered weaker, which lacks sufficient persuasiveness for the experimental results.

4. The experimental evaluation section uses a limited number of benchmarks and fails to include some newer and more comprehensive evaluation benchmarks (MMLU-pro, GPQA, LiveCodeBench, MBPP, HumanEval, MATH-500, Minerva-Math, OlympaidBench, etc.). This limits the ability to fully assess the efficacy and generalizability of the proposed NAIT method.

**Questions:**

Please refer to the weaknesses.

---

> ### Author Response · Authors · 2025-11-25
> **Response to Reviewer 76SK [Part 1/3]**
>
> Q1. The central hypothesis, that the effectiveness of instruction data lies in its ability to activate task-relevant neurons, is merely an assumption and is not widely recognized or proven.
>
> A1. We thank the reviewer for raising this critical question. We fully appreciate your concern that the premise—"the effectiveness of instruction data depends on its ability to activate task-related neurons"—may appear to be a strong theoretical speculation. However, we wish to clarify that this core hypothesis is not merely speculative; it is grounded in existing interpretability literature and, more importantly, is substantiated by our newly added experimental evidence.
>
> (1) Theoretical Foundation
>
> Our hypothesis is predicated on prior research in interpretability [1, 2, 3]. These studies demonstrate that specific neurons within Large Language Models (LLMs) are specialized for processing distinct knowledge or skills. Consequently, we posit that instruction data capable of effectively activating these specific "task neurons" will exhibit greater efficacy during the fine-tuning process.
>
> (2) Empirical Verification (New Experiments)
>
> To further alleviate concerns and translate this hypothesis from theoretical deduction into empirical conclusion, we have conducted a supplementary analysis regarding the Intensity of Neuron Activation (refer to Section 5.1, Page 7 of the revised manuscript). In this experiment, to directly validate whether "the capacity to activate task-related neurons determines the effectiveness of instruction data," we categorized the instruction data into three distinct groups based on their "neuron activation feature alignment scores" for a comparative analysis:
>
> High: The top 10% of data exhibiting the highest neuron activation scores.
> Low: The bottom 10% of data exhibiting the lowest neuron activation scores.
> Random: A randomly sampled 10% subset serving as the baseline.
>
> **Table 3: Performance comparison across datasets at 10% IT dataset under different neuron activation levels.**
> | Intensity | MMLU | BBH | H-Eval | TydiQA | H-Eval | Overall AVG | Overall *Δ (↑)* |
> | :---: | :---: | :---: | :---: | :---: | :---: | :---: | :---: |
> | Random | **47.14** | 39.21 | 14.13 | 44.16 | 25.55 | 34.04 | - |
> | High | 46.83 | **40.02** | **16.53** | **46.09** | **26.44** | **35.18** | +3.35% |
> | Low | 46.30 | 28.15 | 10.61 | 36.12 | 20.15 | 28.27 | -17.54% |
>
> As illustrated in Table 3, the High group outperforms the Random baseline by 3.35%, demonstrating that samples capable of strongly activating relevant neurons effectively enhance model capabilities. More critically, the Low group exhibits a substantial performance degradation of 17.54% compared to the baseline. This result provides direct confirmation of our hypothesis: there exists a strong causal relationship between the effectiveness of data samples and their capacity to activate specific neuronal patterns. Notably, the performance collapse observed in the Low group suggests that data failing to activate (or incorrectly activating) relevant neurons is not merely ineffective but may actively impair the model's inherent capabilities.

---

> ### Author Response · Authors · 2025-11-25
> **Response to Reviewer 76SK [Part 2/3]**
>
> Q2. The use of a small, in-domain reference dataset to extract the neuron activation features (for the target capability) may introduce significant bias into the feature extraction process, potentially limiting the generality of the resulting selection criterion.
>
> A2: We sincerely thank the reviewer for this constructive comment. In fact, to investigate the dependency of NAIT on the size of in-domain data, we have previously conducted relevant experiments in Section 5.1 (Page 8, Figure 3) of the original paper. These experiments employed an incremental testing approach with extremely small sample sizes (e.g., 16, 64, and 256 shots). The results demonstrated that even in the 16-shot scenario, models fine-tuned with data selected by NAIT generally outperformed random selection. Furthermore, when the sample size exceeded 64, the model performance was significantly superior to the random baseline. This provides strong evidence of NAIT's robustness even in scenarios with an extreme scarcity of in-domain data.
>
> To further address the reviewer's concern regarding data sensitivity, we conducted additional experiments for the +NAIT (GSM) setting using three different random seeds for 16-shot sampling. The results are presented in the following table:
>
>
> **Table: Performance comparison of +NAIT (GSM) across different random seeds under the 16-shot setting.**
>
> | Baselines Benchmark→ | MMLU | MMLU-Pro | GSM | SVAMP |  H-Eval |  MBPP |  TydiQA |  XQuAD | BBH | AVG. | $\Delta$ |
> | :--- | :---: | :---: | :---: | :---: | :---: | :---: | :---: | :---: | :---: | :---: | :---: |
> | Alpaca-GPT4 | 46.87 | 21.89 | 14.63 | 39.00 | 27.87 | 51.58 | 39.48 | 42.99 | 39.94 | 36.03 | -
> | Random      | 47.14 | 21.43 | 14.13 | 35.67 | 25.55 | 47.35 | 44.16 | 46.56 | 39.21 | 35.69 | -0.94%
> | +NAIT (GSM)_seed 1  | 46.30 | 21.07 | 17.06 | 45.00 | 27.84 | 48.41 | 41.22 | 44.61 | 39.17 | 36.74 | +1.97%
> | +NAIT (GSM)_seed 2  | 46.81 | 21.89 | 17.13 | 43.00 | 29.27 | 47.88 | 44.13 | 46.85 | 37.59 | 37.17 | +3.16%
> | +NAIT (GSM)_seed 3  | 46.08 | 22.57 | 16.67 | 44.00 | 29.88 | 49.47 | 43.42 | 45.51 | 38.61 | 37.35 | +3.66%
>
> (1) Consistent Improvement in the Target Domain
>
> When GSM is used as the in-domain set, the model achieves significant and consistent performance gains on mathematical reasoning tasks (GSM, SVAMP) regardless of random seed variations. Results across all seeds substantially outperform both full fine-tuning and random selection baselines, with peak performances of 17.13 on GSM8K and 45.00 on SVAMP. This demonstrates that as long as the reference set aligns with the target domain, NAIT can stably identify and enhance domain-specific capabilities.
>
> (2) Robustness in Non-Target Domains
>
> As noted by the reviewer, we indeed observed certain performance fluctuations in non-target tasks (e.g., Multilingual Understanding on TydiQA, with a minimum score of 41.22). These fluctuations reflect the inherent variance characteristics associated with extreme few-shot (16-shot) selection. However, it is crucial to emphasize that despite these local variations, the average performance (AVG) of NAIT remains robust across all seeds, showing an improvement of +1.97% to +3.66% over the baseline. This indicates that while NAIT enhances specific domain capabilities, it maintains an overall robustness superior to the random baseline.
>
>
> Q3. The primary base model used for experimentation is LLaMA and Mistral. Compared to other similarly-sized open-source LLMs (Qwen2.5, Qwen3) available today, LLaMA and Mistral series are generally considered weaker, which lacks sufficient persuasiveness for the experimental results.
>
>
> A3: We fully concur that validating the generalization capability of our method on state-of-the-art open-source models, such as Qwen-2.5 and LLaMA-3, is essential to demonstrate its practical utility and robustness.
>
> To specifically address your concern regarding the currency of the base models, we have incorporated additional experimental results using Qwen-2.5-7B in the Revised Manuscript. As detailed below, NAIT achieved an average score of 75.20, which consistently surpasses both the random selection baseline (73.39) and full fine-tuning (72.42). These findings substantiate that even when applied to base models that already possess strong inherent capabilities, NAIT continues to deliver stable performance gains through the effective selection of high-quality data.
>
> **Table: Effectiveness of Nait Across Models. Alpaca and Random are trained on the full dataset and a random of 10% subset, while Nait selects a 10% subset.**
> |  | | | |Qwen-2.5-7b | | | |
> | :--- | :---: | :---: | :---: | :---: | :---: | :---: | :---: |
> | Method↓ Benchmark→ | MMLU | BBH | GSM | TydiQA | H-Eval| AVG. | $\Delta$ |
> | Alpaca | 73.30 | 65.46 | 83.17 | 49.33 | 90.85 | 72.42 | - |
> | +Random | 74.00 | 67.78 | **84.31** | 51.25 | 89.63 | 73.39 | +1.34% |
> | +Nait | **74.21** | **67.87** | 83.70 | **58.14** | **92.07** | **75.20** | **+3.83%** |

---

> ### Author Response · Authors · 2025-11-25
> **Response to Reviewer 76SK [Part 3/3]**
>
> Q4. The experimental evaluation section uses a limited number of benchmarks and fails to include some newer and more comprehensive evaluation benchmarks (MMLU-pro, GPQA, LiveCodeBench, MBPP, HumanEval, MATH-500, Minerva-Math, OlympaidBench, etc.). This limits the ability to fully assess the efficacy and generalizability of the proposed NAIT method.
>
> A4. We fully agree with the importance of evaluating models on newer and more comprehensive benchmarks to ensure a rigorous assessment of efficacy and generalizability.
>
> To address this concern, we have incorporated four additional benchmarks in the Revised Manuscript, specifically targeting more challenging reasoning scenarios and Out-Of-Distribution (OOD) generalization tests (details provided in Section 4.1, Page 5):
>
> Factual Knowledge: We introduced MMLU-Pro [4], which poses a higher difficulty level than the standard MMLU.
>
> Mathematical Reasoning: We added SVAMP [5] to test OOD generalization capabilities.
>
> Coding Capability: We supplemented our evaluation with MBPP [6].
>
> Multilingual Understanding: We included XQuAD [7].
>
> As presented in Table 2 (Page 6), the experimental results demonstrate that data selected by NAIT based on neuron activation patterns not only performs exceptionally well on original tasks but also enables the model to exhibit superior robustness and generalization in unseen OOD tasks and more difficult scenarios. Specifically, Mathematical Generalization: Using GSM features for selection, NAIT (System 08) achieved a superior score of 41.33 on the SVAMP task, outperforming all other baseline methods. Coding Capability: On MBPP, NAIT (System 09) achieved a high score of 49.74. This result is second only to full-data fine-tuning, indicating that our method maintains highly competitive performance while significantly reducing the data volume required for training.
>
> Reference:
>
> [1] Let’s focus on neuron: Neuron-level supervised fine-tuning for large language model. COLING, 2025.
>
> [2] Unveiling factual recall behaviors of large language models through knowledge neurons. EMNLP, 2024.
>
> [3] Neuron-level knowledge attribution in large language models. EMNLP, 2024.
>
> [4] MMLU-Pro: A More Robust and Challenging Multi-Task Language Understanding Benchmark. NeurIPS, 2024.
>
> [5] Are NLP Models really able to Solve Simple Math Word Problems? NAACL, 2021.
>
> [6] On the cross-lingual transferability of monolingual representations. CoRR, 2021.
>
> [7] Program Synthesis with Large Language Models. CoRR, 2021.

---

> > ### Comment · Reviewer_76SK · 2025-11-28
> >
> > We appreciate the author's explanation and the additional experiments provided.
> > However, I still have concerns regarding the model's OOD capability.
> > To address this, I would suggest completing the following logic-intensive benchmarks on the $\text{Qwen2.5-7B}$ model: LiveCodeBench, MBPP, HumanEval, MATH-500, Minerva-Math, and OlympiadBench.
> > Considering that world knowledge capacity is primarily accumulated during the pre-training phase, demonstrating strong generalization performance on tasks that require advanced logical reasoning would substantially validate the feasibility and robustness of the proposed method.

---

> > > ### Author Response · Authors · 2025-12-04
> > > **Response to Reviewer 76SK [Part1/2]**
> > >
> > > A: We sincerely appreciate the reviewer’s constructive feedback and the suggestion to evaluate our model on logic-intensive benchmarks. We agree that demonstrating strong generalization on OOD tasks—specifically in mathematics and coding—is crucial for validating the robustness of our proposed method.
> > > To address this, we conducted comprehensive evaluations on the recommended and extended benchmarks:
> > > For mathematical reasoning from [1], we use: gsm8k[2], svamp[3], gsm_hard[4], minerva_math[5], math500[6], math[7], asdiv[8], mawps[9]
> > > For Coding Ability from [10] and [11], we use: HumanEval[12], MBPP[13], LiveCodeBench v1[14]. We utilized the standard evaluation harnesses provided in our paper for these datasets to ensure fair comparison. The result as follows:
> > >
> > > 1. Performance on Standard Instruction Tuning Data (Alpaca-GPT4)
> > >
> > > First, we applied our method using the Alpaca-GPT4 dataset as the base. The results are summarized below:
> > >
> > > **Table: Performance comparison of +NAIT (GSM) on 10% selection Alpaca-GPT4 IT Dataset in Qwen**
> > >
> > > | Baselines↓ Benchmark→ | GSM8K | SVAMP |GSM Hard| minerva_math | MATH500 | MATH | ASDIV | MAWPS | AVG. | $\Delta$ |
> > > | :---:                | :---: | :---: | :---: | :---:         | :---: | :---: | :---: | :---: | :---: | :---: |
> > > | Alpaca-GPT4          | 83.17 | 89.67 | 61.30 | 46.00         | 50.80 | 19.40 | 90.90 | 96.70 | 67.20 |   -   |
> > > | Random               | 84.31 | 89.33 | 62.70 | 47.40         | 54.80 | 19.60 | 92.30 | 97.20 | 68.40 | +1.30 |
> > > | +NAIT (GSM)                 | 82.71 | 90.67 | 63.40 | 47.60         | 54.00 | 19.80 | 92.00 | 98.20 | 68.50 | +1.20 |
> > > | +NAIT (Comprehensive)       | 83.70 | 92.00 | 62.70 | 47.80         | 54.00 | 19.20 | 92.00 | 96.90 | 68.50 | +1.30 |
> > >
> > >
> > > **Table: Performance comparison of +NAIT (CodeX) on 10% selection Alpaca-GPT4 IT Dataset**
> > >
> > > | Baselines↓ Benchmark→ |  H-Eval | MBPP | LiveCodeBench | AVG. | $\Delta$ |
> > > | :---:           | :---: | :---: | :---: | :---: | :---: |
> > > | Alpaca-GPT4     | 90.85 | 92.60 | 28.75 | 70.73 |   -   |
> > > | Random          | 89.63 | 92.33 | 33.00 | 71.65 | +0.92 |
> > > | +NAIT (CodeX)          | 93.29 | 93.65 | 35.25 | 74.06 | +3.33 |
> > > | +NAIT (Comprehensive)  | 92.07 | 93.92 | 30.50 | 72.16 | +1.43 |
> > >
> > > Analysis of Initial Results:
> > > As shown in Table 2, our method demonstrates significant improvements in coding capabilities, particularly on the LiveCodeBench, outperforming the baseline and random selection strategies. However, as observed in Table 1, the gains in mathematical reasoning were less pronounced, especially compared with random baseline. We hypothesized that this plateau might be attributed to the inherent strength of the Qwen2.5-7B model and the quality limitations of the Alpaca-GPT4 dataset, which lead to catastrophic overtraining and a performance ceiling of LLM[15,16].

---

> > > ### Author Response · Authors · 2025-12-04
> > > **Response to Reviewer 76SK [Part2/2]**
> > >
> > > 2. Validation on Advanced Instruction Data (Evol-Instruct)
> > >
> > > To verify this hypothesis and further test the OOD capabilities of our method, we conducted a second set of experiments using the higher-quality Evol-Instruct IT dataset[15]. This dataset is known for higher complexity and better logic coverage. The results are presented below:
> > >
> > > **Table: Performance comparison of +NAIT (GSM) on 10% selection Evol-Instruct IT Dataset(70k)**
> > >
> > > | Baselines Benchmark→ | GSM8K | SVAMP |GSM Hard| minerva_math | MATH500 | MATH | ASDIV | MAWPS | AVG. | $\Delta$ |
> > > | :---:                | :---: | :---: | :---:  | :---:        | :---:   | :---: | :---:  | :---: | :---: | :---: |
> > > | Full Evol-Instruct   | 79.80 | 86.80 | 59.70  |  39.20       | 36.60   | 38.50  | 89.50  | 96.20   | 65.80  |   -   |
> > > | Random               | 79.30 | 87.80 | 57.80  |  40.20       | 36.00   | 38.00  | 89.90  | 97.30   | 65.79  | -0.10 |
> > > | +NAIT (GSM)                 | 80.00 | 88.90 | 57.50  |  43.00       | 36.60   | 38.50  | 90.60  | 97.20   | 66.54  | +0.74 |
> > > | +NAIT (Comprehensive)       | 79.20 | 87.90 | 59.70  |  42.20       | 35.60   | 38.00  | 90.30  | 97.50   | 66.30  | +0.50 |
> > >
> > >
> > > **Table: Performance comparison of +NAIT (CodeX) on 10% selection Evol-Instruct IT Dataset(70k)**
> > >
> > > | Baselines Benchmark→ |  H-Eval | MBPP | LiveCodeBench | AVG. | $\Delta$ |
> > > | :---:             | :---: | :---: | :---: | :---: | :---: |
> > > |Full Evol-Instruct | 91.46 | 90.48 | 22.75 | 68.23 |   -
> > > | Random            | 91.86 | 89.31 | 17.75 | 66.31 | -1.92
> > > | +NAIT (CodeX)            | 93.90 | 92.86 | 24.50 | 70.42 | +2.19
> > > | +NAIT (Comprehensive)    | 91.46 | 92.06 | 22.75 | 68.76 | +0.53
> > >
> > > On Math tasks, NAIT (+GSM) achieves 66.54, surpassing both the Full Dataset (65.80) and Random selection (65.79). On Code tasks, NAIT (+CodeX) reaches 70.42, significantly outperforming the Full Dataset (68.23). This indicates that NAIT effectively filters out noise or overly complex but ineffective samples in Evol-Instruct, retaining only the high-quality subset that drives capability improvements. All in all, these OOD experiments confirm that NAIT's effectiveness is not limited to specific datasets.
> > >
> > >
> > >
> > > [1] https://github.com/ZubinGou/math-evaluation-harness.
> > >
> > > [2] Training Verifiers to Solve Math Word Problems, CoRR, 2021.
> > >
> > > [3] Are NLP Models really able to Solve Simple Math Word Problems? NAACL, 2021.
> > >
> > > [4] PAL: Program-aided Language Models. ICML, 2023.
> > >
> > > [5] Solving Quantitative Reasoning Problems with Language Models. NeurIPS 2022.
> > >
> > > [6] https://huggingface.co/datasets/HuggingFaceH4/MATH-500.
> > >
> > > [7] Measuring Mathematical Problem Solving With the MATH Dataset. NeurIPS, 2021.
> > >
> > > [8] A Diverse Corpus for Evaluating and Developing English Math Word Problem Solvers. ACL, 2020.
> > >
> > > [9] MAWPS: A Math Word Problem Repository. NAACL, 2016.
> > >
> > > [10] https://github.com/evalplus/evalplus.
> > >
> > > [11] https://github.com/huggingface/lighteval.
> > >
> > > [12] Evaluating Large Language Models Trained on Code. CoRR, 2021.
> > >
> > > [13] On the cross-lingual transferability of monolingual representations. CoRR, 2021.
> > >
> > > [14] LiveCodeBench: Holistic and Contamination Free Evaluation of Large Language Models for Code. ICLR, 2025.
> > >
> > > [15] Overtrained Language Models Are Harder to Fine-Tune. ICLR, 2025.
> > >
> > > [16] Limitations of refinement methods for weak to strong generalization. CoLM, 2025.
> > >
> > > [17] WizardLM: Empowering Large Pre-Trained Language Models to Follow Complex Instructions. ICLR, 2024.

---

### Official Review · Reviewer_SZW9 · 2025-10-30

**Soundness:** 3
**Presentation:** 3
**Contribution:** 3
**Rating:** 4
**Confidence:** 4

**Summary:**

This paper introduces NAIT, a novel framework for efficient instruction tuning data selection by analyzing neuron activation patterns in Large Language Models to identify data that effectively enhances specific or general capabilities. The core hypothesis is that IT data samples are more effective if their neuronal activation patterns closely align with the activation characteristics associated with a target capability. NAIT extracts reusable neuron activation features from in-domain datasets and selects optimal IT samples based on their similarity to these target capability activation features, demonstrating superior performance and cost efficiency compared to existing methods. Experiments show NAIT consistently improves LLM performance (e.g., 3% with only 10% of data), reveals the strong general transferability of logical reasoning and programmatic features, and identifies a stable core subset of data universally beneficial across diverse tasks. Ablation studies reveal that optimal performance is achieved with approximately 30% of the IT dataset selected by NAIT, with larger proportions leading to degradation due to redundant data.  The paper open-sources a cross-task neuron feature library and the Alpaca-NAIT dataset, a high-quality IT dataset curated using the proposed NAIT framework.

**Strengths:**

1. The idea of using the model’s own internal neuron activation patterns to evaluate data quality is novel and interesting. Unlike prior data selection methods that rely on external scorers or surface heuristics, NAIT directly taps into the model’s internal representation. This provides a more interpretable and fine-grained signal and is externally independent.
2. NAIT effectively addresses the scenario of enhancing specific capabilities in an LLM. The method is flexible – by providing an in-domain sample set for a desired skill, the framework can pick out relevant training data to boost that skill. Experiments confirm that NAIT tuning on a skill-focused subset yields better performance on that domain’s benchmark than generic fine-tuning.
3. The approach shows that a 10% subset selected by NAIT can outperform or equal the full 100% data training in many cases. These results reinforce the claim that more data isn’t always better, and smart selection can yield efficiency gains. It’s impressive that NAIT even beat methods like AlpaGasus and Q2Q which leverage GPT4 or uncertainty measures.
4. A notable strength is the paper’s analysis of neuron activation transferability. Those findings provide insight into what kinds of instructional data are broadly useful. Such analysis adds depth to the work, beyond just performance numbers.
5. By avoiding external model calls and using efficient feature extraction, NAIT is relatively lightweight. The authors compare computational cost and show significant speedup and cost reduction versus prior methods.

**Weaknesses:**

1. A practical concern is that NAIT requires a representative in-domain dataset for each target capability as a starting point. In real scenarios, such labeled data may not be readily available for every “capability” one wishes to improve. This reliance potentially limits NAIT’s applicability to cases where one can clearly define and obtain data for the target skill. If the in-domain examples are too few or not truly representative, the quality of the activation feature (and thus the selection) might suffer. The paper does not deeply explore how sensitive the method is to the choice or size of this in-domain set.
2. One of the claims is that NAIT can enhance specific or general capabilities. However, the results suggest that focusing on a single strong capability (like GSM for math) yielded the highest average boost, even more than combining all capability features. The model fine-tuned on the GSM-selected subset outperformed the model using a multi-capability combined subset (35.42 vs 35.18 AVG). This is somewhat counter-intuitive – ideally, integrating multiple skills would give broader improvement. The fact that the “all capabilities” selection did not significantly outperform single-capability selections raises concerns. It suggests NAIT might struggle to simultaneously optimize for many skills, or that the method for merging activation features may be suboptimal. This limitation should be better addressed: if one wants a generally strong model, do they have to run NAIT for every capability and merge data ad-hoc?
3. The paper lacks direct comparison to an embedding-based selection baseline. How does NAIT compare to a simpler strategy like using embedding similarity for data selection? For instance, one could embed each candidate instruction (using the same LLM) and each in-domain example, then select those with highest similarity to the in-domain set (or to a mean in-domain embedding). This makes it harder to assess how much benefit comes from the neuron-level analysis versus just using the model’s last-layer representation or other heuristics.
4. There are some open questions about NAIT’s scalability and certain design choices. The paper would benefit from clarity on which layers or neurons are used for the activation pattern – is it the entire hidden state of the model, a subset of layers, or aggregated over the whole sequence? Additionally, while the authors claim adaptability to different model sizes, it’s not explicitly shown how NAIT scales to very large models (e.g., 70B parameters) – the experiments seem limited to one model family (likely around 7B-13B scale).

**Questions:**

1. Could you clarify the construction of the “activation feature” and alignment score? Specifically: (a) Do you use the entire model’s neuron activations for the input (all layers) or only a particular layer’s output (e.g., last hidden state)? (b) What is ∆A($P_i$) exactly (activation difference relative to what baseline)? (c) Once you have PCA-compressed features for in-domain and candidate data, is the similarity score simply a dot product or cosine similarity in that feature space? Providing these details (perhaps I missed them in the text).
2. In the experiment combining all capability features, how was this combination done? Did you union the top-k selections from each capability run, or aggregate the activation features into one composite feature vector for scoring all data at once? The result that this combined approach didn’t outperform the best single-capability selection is interesting – could you elaborate on why you think that happened?
3. Have you tried applying NAIT with a larger base model or on a larger pool of candidate data? The cost analysis suggests NAIT is efficient on 52k data with a 7B-ish model. If we consider, say, a 70B model and a few hundred thousand instructions, do any new challenges arise (e.g., needing to handle many more neurons, longer activation vectors, etc.)?

---

> ### Author Response · Authors · 2025-11-25
> **Response to Reviewer SZW9 [Part 1/5]**
>
> Q1. A practical concern is that NAIT requires a representative in-domain dataset for each target capability as a starting point. In real scenarios, such labeled data may not be readily available for every “capability” one wishes to improve. This reliance potentially limits NAIT’s applicability to cases where one can clearly define and obtain data for the target skill. If the in-domain examples are too few or not truly representative, the quality of the activation feature (and thus the selection) might suffer. The paper does not deeply explore how sensitive the method is to the choice or size of this in-domain set.
>
> A1. We sincerely thank the reviewer for this constructive comment. In fact, to investigate the dependency of NAIT on the size of in-domain data, we have previously conducted relevant experiments in Section 5.1 (Page 8, Figure 3) of the original paper. These experiments employed an incremental testing approach with extremely small sample sizes (e.g., 16, 64, and 256 shots). The results demonstrated that even in the 16-shot scenario, models fine-tuned with data selected by NAIT generally outperformed random selection. Furthermore, when the sample size exceeded 64, the model performance was significantly superior to the random baseline. This provides strong evidence of NAIT's robustness even in scenarios with an extreme scarcity of in-domain data.
>
> To further address the reviewer's concern regarding data sensitivity, we conducted additional experiments for the +NAIT (GSM) setting using three different random seeds for 16-shot sampling. The results are presented in the following table:
>
>
> **Table: Performance comparison of +NAIT (GSM) across different random seeds under the 16-shot setting.**
>
> | Baselines Benchmark→ | MMLU | MMLU-Pro | GSM | SVAMP |  H-Eval |  MBPP |  TydiQA |  XQuAD | BBH | AVG. | $\Delta$ |
> | :--- | :---: | :---: | :---: | :---: | :---: | :---: | :---: | :---: | :---: | :---: | :---: |
> | Alpaca-GPT4 | 46.87 | 21.89 | 14.63 | 39.00 | 27.87 | 51.58 | 39.48 | 42.99 | 39.94 | 36.03 | -
> | Random      | 47.14 | 21.43 | 14.13 | 35.67 | 25.55 | 47.35 | 44.16 | 46.56 | 39.21 | 35.69 | -0.94%
> | +NAIT (GSM)_seed 1  | 46.30 | 21.07 | 17.06 | 45.00 | 27.84 | 48.41 | 41.22 | 44.61 | 39.17 | 36.74 | +1.97%
> | +NAIT (GSM)_seed 2  | 46.81 | 21.89 | 17.13 | 43.00 | 29.27 | 47.88 | 44.13 | 46.85 | 37.59 | 37.17 | +3.16%
> | +NAIT (GSM)_seed 3  | 46.08 | 22.57 | 16.67 | 44.00 | 29.88 | 49.47 | 43.42 | 45.51 | 38.61 | 37.35 | +3.66%
>
> (1) Consistent Improvement in the Target Domain
>
> When GSM is used as the in-domain set, the model achieves significant and consistent performance gains on mathematical reasoning tasks (GSM, SVAMP) regardless of random seed variations. Results across all seeds substantially outperform both full fine-tuning and random selection baselines, with peak performances of 17.13 on GSM8K and 45.00 on SVAMP. This demonstrates that as long as the reference set aligns with the target domain, NAIT can stably identify and enhance domain-specific capabilities.
>
> (2) Robustness in Non-Target Domains
>
> As noted by the reviewer, we indeed observed certain performance fluctuations in non-target tasks (e.g., Multilingual Understanding on TydiQA, with a minimum score of 41.22). These fluctuations reflect the inherent variance characteristics associated with extreme few-shot (16-shot) selection. However, it is crucial to emphasize that despite these local variations, the average performance (AVG) of NAIT remains robust across all seeds, showing an improvement of +1.97% to +3.66% over the baseline. This indicates that while NAIT enhances specific domain capabilities, it maintains an overall robustness superior to the random baseline.

---

> ### Author Response · Authors · 2025-11-25
> **Response to Reviewer SZW9 [Part 2/5]**
>
> Q2. One of the claims is that NAIT can enhance specific or general capabilities. However, the results suggest that focusing on a single strong capability (like GSM for math) yielded the highest average boost, even more than combining all capability features. The model fine-tuned on the GSM-selected subset outperformed the model using a multi-capability combined subset (35.42 vs 35.18 AVG). This is somewhat counter-intuitive – ideally, integrating multiple skills would give broader improvement. The fact that the “all capabilities” selection did not significantly outperform single-capability selections raises concerns. It suggests NAIT might struggle to simultaneously optimize for many skills, or that the method for merging activation features may be suboptimal. This limitation should be better addressed: if one wants a generally strong model, do they have to run NAIT for every capability and merge data ad-hoc?
>
>
> A2: We sincerely appreciate the reviewer's profound insight regarding the trade-off between single-capability and multi-capability selection. We indeed observed that the model fine-tuned on a subset selected based on GSM (System 08) outperformed the multi-capability combined model based on feature averaging (System 12) in terms of average performance. To address this seemingly "counter-intuitive" phenomenon and your subsequent concerns, we provide the following explanation and proposed solution:
>
> (1) Theoretical Explanation: Feature Interference and Vector Dilution
>
> As demonstrated in our visualization analysis (Figure 6, Section 5.3, Page 9), the activation directions corresponding to different capabilities (e.g., mathematical reasoning vs. programming) are often non-overlapping within the high-dimensional representation space. The strategy adopted by System 12 involves Feature Averaging of these distinct feature vectors. This causes the resulting feature vector to point towards the geometric center, leading to a phenomenon of "vector dilution." As observed by Dong et al. (2024) [1], significant "performance conflicts" exist when mixing different capabilities. Forcing the fusion of screening signals from multiple objectives at the feature level weakens the directionality of the final vector on specific high-difficulty tasks (such as GSM). This explains why the targeted GSM selection (System 08) yields sharper improvements compared to feature averaging (System 12).
>
> (2) Shifting from "Feature Averaging" to "Data Union"
>
> You raised a critical question: Is it necessary to run NAIT for every capability and merge the data ad-hoc? To investigate this, we conducted an in-depth analysis of the data distribution selected by different capabilities. As shown in Figure 7, Page 10, there is a 60.21% overlap among datasets selected by different activation capabilities. This indicates the existence of a stable "General Core" data subset that is commonly activated by various high-intelligence tasks (e.g., reasoning, coding).
>
> Consequently, we conducted a new experiment: instead of averaging the feature vectors, we ran NAIT separately to select Top-k data for each capability and then fine-tuned the model on the Union of these data subsets. The results are presented in the table below:
>
> | Random Seed Benchmark→ | MMLU | MMLU-Pro | GSM | SVAMP |  H-Eval |  MBPP |  TydiQA |  XQuAD | BBH | AVG. | $\Delta$ |
> | :--- | :---: | :---: | :---: | :---: | :---: | :---: | :---: | :---: | :---: | :---: | :---: |
> | Alpaca-GPT4 | 46.87 | 21.89 | 14.63 | 39.00 | 27.87 | 51.58 | 39.48 | 42.99 | 39.94 | 36.03 | -
> | NAIT (8-12) | 46.83 | 23.29 | 16.53 | 39.67  |26.44 | 47.62 | 46.09 | 48.27 | 40.02 | 37.20 | +3.24%
> | Union NAIT (8-12) |  47.07 | 22.39 | 16.38 | 39.33 | 31.1 | 48.94 | 48.51 | 49.23 | 39.72  | 38.07 | + 5.35%
>
> Note: Union NAIT (Data Union) refers to the result of merging and de-duplicating the data subsets selected in Systems 07-11.
>
> The results demonstrate that adopting the Data Union strategy allows the model to maintain high performance on specific tasks like GSM (16.38 vs. Baseline 14.63) while achieving significant improvements in coding (H-Eval 31.10) and multilingual tasks (TydiQA 48.51).
>
> In summary, regarding your concern about "whether it is necessary to run NAIT on all capabilities," our response is: while theoretically aggregating signals from multiple capabilities is necessary to achieve an ideal general model, in practice, running NAIT on every potential capability one by one is cost-prohibitive and inefficient. Based on our experimental findings, the optimal trade-off is to select a small number of orthogonal and high-value capabilities (such as Logical Reasoning [GSM] and Programming [CodeX], which show strong transferability as analyzed in Section 5.3, Page 9).

---

> ### Author Response · Authors · 2025-11-25
> **Response to Reviewer SZW9 [Part 3/5]**
>
> Q3. The paper lacks direct comparison to an embedding-based selection baseline. How does NAIT compare to a simpler strategy like using embedding similarity for data selection? For instance, one could embed each candidate instruction (using the same LLM) and each in-domain example, then select those with highest similarity to the in-domain set (or to a mean in-domain embedding). This makes it harder to assess how much benefit comes from the neuron-level analysis versus just using the model’s last-layer representation or other heuristics.
>
> A3.  We fully concur that comparing NAIT against simple selection strategies based on Embedding or Representation [2] is essential to intuitively quantify the gains offered by neuron-level analysis. Accordingly, we have incorporated Embedding-based and Representation-based baseline experiments in the revised paper. Furthermore, to provide a more comprehensive assessment of NAIT's positioning, we additionally introduced LESS [3], a strong gradient-based baseline, for comparison. The detailed experimental results are presented in the table below (and in Appendix E of the revised manuscript):
>
> | Method | MMLU | MMLU-Pro | TydiQA | XQuAD | BBH | AVG |
> | :--- | :---: | :---: | :---: | :---: | :---: | :---: |
> | *+MMLU* | | | | | | |
> | Embedding | 46.29 | 21.07 | 42.97 | 44.73 | 36.67 | 38.35 |
> | Representation-based | 45.57 | 23.0 | 46.78 | 46.85 | 38.15 | 40.07 |
> | LESS | **48.12** | **24.50** | 43.42 | 45.51 | 38.15 | 39.94 |
> | Nait | 47.81 | 23.61 | **47.16** | **49.47** | **38.52** | **41.31** |
> | *+TydiQA* | | | | | | |
> | Embedding-based | **47.49** | 22.54 | 42.4 | 46.53 | 38.43 | 39.48 |
> | Representation-based | 45.91 | 21.04 | 40.02 | 46.0 | 36.02 | 37.80 |
> | LESS | 44.68 | 22.21 | 45.24 | 47.67 | 37.69 | 39.50 |
> | Nait | 46.17 | **22.82** | **47.78** | **49.23** | **40.00** | **41.20** |
> | *+BBH* | | | | | | |
> | Embedding | 47.51 | 22.86 | 43.25 | 46.66 | 36.76 | 39.41 |
> | Representation-based | 45.81 | 21.11 | 36.78 | 44.61 | 37.59 | 37.18 |
> | LESS | 46.85 | **23.75** | **46.56** | 47.67 | 39.72 | 39.61 |
> | Nait | **47.78** | 23.36 | 45.93 | **48.46** | **40.46** | **41.20** |
>
> (1) Comparison with Simple Baselines (Vs. Embedding/Representation)
>
> The experimental results indicate that while Embedding-based and Representation-based methods yield marginal improvements by filtering data via semantic similarity, they significantly underperform compared to NAIT. For instance, when targeting MMLU, NAIT achieves an average score of 41.31, whereas the Embedding method reaches only 38.35. This compellingly demonstrates that relying solely on surface-level semantic similarity or final-layer representations fails to capture complex, task-specific capability features effectively. NAIT successfully bridges this gap by leveraging in-depth neuron-level analysis.
>
> (2) Generalization Advantage over Gradient-based Methods (Vs. LESS)
>
> While LESS, as a gradient-based method, achieves high scores on specific target tasks (e.g., MMLU), it tends to overfit the target validation set, resulting in a substantial decline in transferability to non-target tasks. For example, in the MMLU setting, LESS scores only 38.15 on BBH, which is lower than NAIT's 38.52. Consequently, the overall average performance (AVG) of LESS consistently falls short of NAIT.
>
> In conclusion, NAIT achieves superior overall average performance (AVG) across all settings (Targeting MMLU, TydiQA, BBH). Moreover, as illustrated in Table 5, Section 5.2 of the main text, NAIT circumvents the prohibitive computational overhead associated with gradient calculations required by LESS. This indicates that NAIT strikes an optimal balance among capturing deep capability features, cross-task generalization, and computational efficiency.

---

> ### Author Response · Authors · 2025-11-25
> **Response to Reviewer SZW9 [Part 4/5]**
>
> Q4. There are some open questions about NAIT’s scalability and certain design choices. The paper would benefit from clarity on which layers or neurons are used for the activation pattern – is it the entire hidden state of the model, a subset of layers, or aggregated over the whole sequence? Additionally, while the authors claim adaptability to different model sizes, it’s not explicitly shown how NAIT scales to very large models (e.g., 70B parameters) – the experiments seem limited to one model family (likely around 7B-13B scale).
>
>
> A4. (1) Regarding the activation mode
> In Section 3.2.1 (Page 4, line 206) of the original manuscript, we explicitly specified the location for neuron selection: "we record the activation states of all $J$ neurons in Model $M$ token $t_k$ at the decoder layer."
>
> We selected the decoder layer based on existing research principles, which demonstrate that neurons in the decoder layer effectively aggregate information processed by both the Self-Attention and Feed-Forward Network (FNN) modules, thus providing a comprehensive representation of the model's internal state.
>
> (2) Scalability and Model Diversity
>
> Parameter Scaling (7B to 13B): As noted, in addition to the original Llama-2-7B experiments, we included comprehensive experiments on Llama-2-13B in the original manuscript. The results demonstrate that NAIT maintains significant performance gains on the 13B model, validating the method's effectiveness even when the parameter count is doubled.
> Model Diversity (Qwen-2.5-7B): We have further supplemented our evaluation with results on the state-of-the-art Qwen-2.5-7B model. Experiments show that NAIT achieves an average score of 75.20, consistently outperforming Random Selection (73.39) and Full Fine-tuning (72.42). These results consistently demonstrate that even when the base model possesses strong inherent capabilities, NAIT can still yield stable performance improvements by selecting high-quality data.
>
> (3) Regarding the 70B Model
>
> We are very eager to verify our method on the 70B model; however, due to computational resource constraints, we are currently unable to conduct experiments at this scale. Nevertheless, given the high consistency of NAIT's performance across Llama-2-7B, Llama-2-13B, and the newer Qwen-2.5-7B, we are confident that NAIT will remain effective on larger-scale models.
>
>
> **Table: Effectiveness of Nait Across Models. Alpaca and Random are trained on the full dataset and a random of 10% subset, while Nait selects a 10% subset.**
> |  | | | |Qwen-2.5-7b | | | |
> | :--- | :---: | :---: | :---: | :---: | :---: | :---: | :---: |
> | Method↓ Benchmark→ | MMLU | BBH | GSM | TydiQA | H-Eval| AVG. | $\Delta$ |
> | Alpaca | 73.30 | 65.46 | 83.17 | 49.33 | 90.85 | 72.42 | - |
> | +Random | 74.00 | 67.78 | **84.31** | 51.25 | 89.63 | 73.39 | +1.34% |
> | +Nait | **74.21** | **67.87** | 83.70 | **58.14** | **92.07** | **75.20** | **+3.83%** |

---

> ### Author Response · Authors · 2025-11-25
> **Response to Reviewer SZW9 [Part 5/5]**
>
> Q5. Could you clarify the construction of the “activation feature” and alignment score? Specifically: (a) Do you use the entire model’s neuron activations for the input (all layers) or only a particular layer’s output (e.g., last hidden state)? (b) What is ∆A( ) exactly (activation difference relative to what baseline)? (c) Once you have PCA-compressed features for in-domain and candidate data, is the similarity score simply a dot product or cosine similarity in that feature space? Providing these details (perhaps I missed them in the text).
>
> A5. To make the calculation process more transparent, we have added a detailed Algorithm box in Appendix F of the revised paper. We provide specific clarifications to your questions below:
>
> (1) Layers Used for Activation
>
> As mentioned on Page 4, line 206, we utilize the neuron activations of all decoder layers, not just the final hidden state. As described in Section 3.2.1, for a specific layer $l$ and token $t_k$, we record its activation vector $A(t_k)$. The final alignment score $s_y$ is calculated by summing the projections across all layers $l=1, \dots, L$ (see Equation 5). This ensures that we capture processing patterns throughout the entire depth of the network.
>
> (2) Definition of $\Delta A$
>
> $\Delta A$ represents the dynamic activation shift of a specific layer during the generation process. As defined in Equation 2 (Section 3.2.1), $A^{(l)}(t_K)$ corresponds to the activation vector of the $K$-th token, and $A^{(l)}(t_1)$ corresponds to the activation vector of the start token. We compute this difference to capture the evolution of neuron activation states from the beginning to the end of the sequence, rather than comparing them against a static baseline.
>
> (3) Calculation of Similarity Score
>
> The similarity score is calculated based on projection (dot product), not cosine similarity.
> First, we use PCA on the difference set $\Delta A^{(l)}$ to extract the principal direction vector $v_l$ representing the target capability (Equation 3). Then, for a candidate sample $y$, we project its activation $A^{(l)}$ onto this target direction $v_l$. The final score $s_y$ is the sum of these dot products across all layers. We will revise Section 3.2 in the final version to ensure these definitions are stated even more clearly.
>
>
> Q6: Have you tried applying NAIT with a larger base model or on a larger pool of candidate data? The cost analysis suggests NAIT is efficient on 52k data with a 7B-ish model. If we consider, say, a 70B model and a few hundred thousand instructions, do any new challenges arise (e.g., needing to handle many more neurons, longer activation vectors, etc.)?
>
>
> A6: (1) Scalability on Larger Datasets
>
> In fact, as discussed in Section 5.1 (Page 8, "Different Instruction Tuning Datasets") of the original paper, our experiments were not limited to the Alpaca-52k dataset. We successfully applied NAIT to the Evol-Instruct IT Dataset (70k) [4] and the Oral-GPT4 IT Dataset (100k) [5].
>
> The results on these larger datasets demonstrate that NAIT remains highly efficient as the size of the candidate pool increases. Notably, NAIT is not only scalable but also sensitive to data quality. For instance, it successfully identified that the Evol-Instruct dataset contains a higher proportion of high-quality samples, validating the superior quality of that dataset.
>
> (2) Scalability to Larger Models (e.g., 70B)
>
> Space Complexity: NAIT calculates metrics in batches. Therefore, we can adapt to larger instruction tuning datasets and the larger 70B model simply by reducing the Batch Size.
>
> Computational Complexity: Although the hidden layer dimension increases in a 70B model, NAIT's computation primarily involves vector dot products. The complexity of this operation grows linearly with the dimension. Consequently, the computational overhead remains predictable and acceptable.
>
> Reference:
>
> [1] How Abilities in Large Language Models are Affected by Supervised Fine-tuning Data Composition. ACL, 2024.
>
> [2] Evaluation of similarity-based explanations. In 9th International Conference on Learning Representation. ICLR, 2021.
>
> [3] LESS: selecting influential data for targeted instruction tuning. ICML, 2024.
>
> [4] WizardLM: Empowering Large Pre-Trained Language Models to Follow Complex Instructions. ICLR, 2024.
>
> [5] Orca: Progressive Learning from Complex Explanation Traces of GPT-4. CoRR, 2023.

---

### Official Review · Reviewer_wy8Y · 2025-11-01

**Soundness:** 3
**Presentation:** 2
**Contribution:** 3
**Rating:** 6
**Confidence:** 3

**Summary:**

This paper introduces a neuron-aware framework for selecting instruction-tuning data based on neuron activation similarity. Instead of relying on external models or uncertainty metrics, NAIT extracts neuron activation features from small in-domain reference sets and scores candidate samples by alignment with these features. Experiments  show that training on 10% of the selected Alpaca-GPT-4 data improves average performance across multiple benchmarks and domains.

**Strengths:**

- Strong data‑efficiency. NAIT with 10% of Alpaca‑GPT‑4 matches/exceeds full‑data tuning and outperforms several selection baselines.
- The neuron‑feature construction and per‑candidate alignment score are conceptually simple and effective across many tasks.

**Weaknesses:**

- The experiments didn’t compare against some strong recent data selection methods like LESS (ICML 2024).
- The paper seems doesn’t specify which layers’ activations are used to compute features or why. Since different layers capture different types of information, this choice may matters. explanation or ablations could be added.

**Questions:**

Did you compute ΔA(Dn) analogously to Eq. (3), then apply the same PCA transform learned on in‑domain features before scoring? Do you use cosine similarity or dot product? Any per‑sample length or norm? Providing an algorithm box would be better.

---

> ### Author Response · Authors · 2025-11-25
> **Response to Reviewer wy8Y [Part 1/2]**
>
> Q1. The experiments didn’t compare against some strong recent data selection methods like LESS (ICML 2024).
>
> A1. We fully recognize the significance of gradient-based subset selection methods, such as LESS [1], in the field of sample importance estimation. However,  we would like to first clarify why we initially did not consider such methods: such methods typically necessitate additional backpropagation steps to compute gradient signals, resulting in significantly higher computational costs compared to schemes based solely on forward features. Given that this paper focuses primarily on efficient data selection within resource-constrained scenarios, we initially prioritized baselines based on forward activations, as they offer comparability while avoiding the overhead of gradient computation.
>
> Nevertheless, we find the reviewer’s insight highly valuable. Consequently, to provide a comprehensive evaluation, we have incorporated comparisons with the gradient-based method LESS [1], as well as Embedding-based and Representation-based [2] approaches. The updated results are presented in the table below:
>
> (1) Performance & Generalization
>
> Experimental results indicate that while LESS achieves high scores on specific target tasks (e.g., MMLU), it exhibits a tendency to overfit the target validation set, resulting in suboptimal generalization on non-target tasks. In contrast, NAIT demonstrates superior cross-task transferability while maintaining competitive performance on target tasks. Consequently, NAIT outperforms LESS in terms of overall average performance (e.g., when targeting MMLU, the average score for LESS is 39.94, whereas NAIT achieves 41.31).
>
> (2) Computational Efficiency
>
> We have added a detailed cost comparison on page 9, section 5.1 table 5 (also see below). The results demonstrate that NAIT successfully circumvents the high overhead associated with gradient computation (requiring no backpropagation). It achieves the best overall average performance while significantly reducing computational costs. This further validates NAIT's ability to strike a superior balance between selection efficiency and the capture of deep model capabilities in resource-constrained scenarios.
>
> **Table: Performance comparison of baselines of activating targeted ability using 10% of the IT data. The detailed experiment setting shown on Appendix E, Page 17**
>
> | Method | MMLU | MMLU-Pro | TydiQA | XQuAD | BBH | AVG |
> | :--- | :---: | :---: | :---: | :---: | :---: | :---: |
> | *+MMLU* | | | | | | |
> | Embedding | 46.29 | 21.07 | 42.97 | 44.73 | 36.67 | 38.35 |
> | Representation-based | 45.57 | 23.0 | 46.78 | 46.85 | 38.15 | 40.07 |
> | LESS | **48.12** | **24.50** | 43.42 | 45.51 | 38.15 | 39.94 |
> | Nait | 47.81 | 23.61 | **47.16** | **49.47** | **38.52** | **41.31** |
> | *+TydiQA* | | | | | | |
> | Embedding-based | **47.49** | 22.54 | 42.4 | 46.53 | 38.43 | 39.48 |
> | Representation-based | 45.91 | 21.04 | 40.02 | 46.0 | 36.02 | 37.80 |
> | LESS | 44.68 | 22.21 | 45.24 | 47.67 | 37.69 | 39.50 |
> | Nait | 46.17 | **22.82** | **47.78** | **49.23** | **40.00** | **41.20** |
> | *+BBH* | | | | | | |
> | Embedding | 47.51 | 22.86 | 43.25 | 46.66 | 36.76 | 39.41 |
> | Representation-based | 45.81 | 21.11 | 36.78 | 44.61 | 37.59 | 37.18 |
> | LESS | 46.85 | **23.75** | **46.56** | 47.67 | 39.72 | 39.61 |
> | Nait | **47.78** | 23.36 | 45.93 | **48.46** | **40.46** | **41.20** |
>
> **Table: Efficiency Comparison of Different Methods on NVIDIA A800 80GB with Batch Size set to 8**
>
> > **Note:** API costs follow the official OpenAI pricing ($2.00/million input tokens, $8/million output tokens for GPT-4.1); GPU costs are estimated based on the Google Cloud pricing ($1.15 per GPU hour for NVIDIA A800 80GB).
>
> | Method | Externally-Independent | Time | Cost |
> | :--- | :---: | :---: | :---: |
> | AlpaGasus | ✗ | 19.07h | $178.02 |
> | Q2Q | ✗ | 3.52h | $4.05 |
> | LESS | ✓ | 9.86h | $11.33 |
> | SelectIT | ✓ | 23.20h | $26.68 |
> | **NAIT** | ✓ | **1.32h** | **$1.52** |

---

> ### Author Response · Authors · 2025-11-25
> **Response to Reviewer wy8Y [Part 2/2]**
>
> Q2. The paper seems doesn’t specify which layers’ activations are used to compute features or why. Since different layers capture different types of information, this choice may matters. explanation or ablations could be added.
>
>
> A2. In fact, the specific location for neuron selection was explicitly stated in Section 3.21 (Page 4) of our original manuscript: "we record the activation states of all $J$ neurons in Model $M$ token $t_k$ at the decoder layer." This design choice is primarily motivated by the following two considerations:
>
> First, from an architectural perspective, the output state of each layer in the Decoder Block is essentially a linear superposition of the input state and the transformations processed by the Self-Attention and Feed-Forward Network (FFN) modules. This implies that the output of any given layer effectively aggregates the cumulative information generated by that layer and all preceding ones.
>
> Second, we posit that manually selecting specific layers risks the loss of critical information. Recent research in Mechanistic Interpretability indicates that Large Language Models (LLMs) exhibit a widespread Superposition phenomenon [3,4]. This suggests that features are not exclusively encoded by single neurons but are instead distributed across the high-dimensional space in the form of complex linear combinations. Furthermore, complex semantic reasoning often relies on the synergistic operation of cross-layer Circuits [5]. Consequently, treating the model as a unified system and utilizing the full states of all layers allows for the maximal preservation of distributed features and deep semantic dependencies within the model, thereby yielding the most robust performance.
>
>
>
> Q3. Did you compute ΔA(Dn) analogously to Eq. (3), then apply the same PCA transform learned on in‑domain features before scoring? Do you use cosine similarity or dot product? Any per‑sample length or norm? Providing an algorithm box would be better.
>
> A3. We have added a detailed Algorithm box in Appendix F on Page 19 of the revised manuscript to make the entire calculation process more transparent. Addressing your specific questions, we clarify the following:
>
> (1) Regarding feature calculation for candidate data
>
> For the Candidate Dataset ($D_{ins}$), we do not calculate the start-end activation difference $\Delta A$ (as shown in Eq. 2) used for In-domain data. Instead, as shown in Lines 23-24 of Algorithm 1, we calculate the Mean Activation of all tokens in the sequence, denoted as $A^{(l)}$.
> Reason: In the feature extraction phase (Stage 1), $\Delta A$ is used to capture the capability direction when a specific capability is triggered. However, in the data filtering phase (Stage 2), our goal is to identify samples that continuously and strongly activate these specific neuronal directions throughout the generation process. Therefore, matching the sample's mean activation with the target direction is more appropriate.
>
> (2) Regarding PCA transformation
>
> We do not re-perform PCA on the IT data. We directly utilize the fixed direction vectors $v_l$ extracted exclusively from the In-domain dataset $P$ during Stage 1. We project the mean activation values of the candidate samples onto these pre-computed directions.
>
> (3) Dot Product vs. Cosine Similarity
>
> We employ the Dot Product. As indicated in Eq. (5) of the main text ($s_y = \sum_{l=1}^{L} (A^{(l)} \cdot v_l)$) and Line 24 of Algorithm 1, the score $s_y$ is the sum of the dot products between the candidate sample's activation vector and the target direction vector. This approach accounts for both directional alignment and the preservation of activation magnitude information.
>
> (4) Regarding sample length or Normalization
>
> To handle varying sequence lengths, we apply Mean Pooling over the activation values of all tokens in the sequence to obtain the activation vector $A^{(l)}$ for each layer. This naturally normalizes the impact of inconsistent sequence lengths across different samples.
>
> Reference:
>
> [1] LESS: selecting influential data for targeted instruction tuning. ICML, 2024.
>
> [2] Evaluation of similarity-based explanations. In 9th International Conference on Learning Representation. ICLR, 2021.
>
> [3] Towards Monosemanticity: Decomposing Language Models With Dictionary Learning. Anthropic, 2023.
>
> [4] Sparse Autoencoders Find Highly Interpretable Features in Language Models. ICLR, 2024.
>
> [5] Interpretability in the Wild: a Circuit for Indirect Object Identification. ICLR, 2023.

---

### Official Review · Reviewer_2hWE · 2025-11-02

**Soundness:** 3
**Presentation:** 3
**Contribution:** 2
**Rating:** 4
**Confidence:** 4

**Summary:**

- This paper introduces NAIT, a novel framework for selecting high-quality data for the instruction tuning LLMs.
- The proposed NAIT method first captures a characteristic activation feature for a target capability by processing a small set of examples and applying PCA to their neuron activations. Second, it scores and selects data from a larger candidate pool by measuring the alignment between the candidate data's neuron activations and this characteristic feature.

**Strengths:**

- While neuron activation analysis has been used to interpret LLMs, its application as a direct signal for instruction data selection is novel and interpretable.
- The paper is well written, and the experiments are conducted on a wide range of datasets and base models.
- The cost-efficiency analysis is a crucial and compelling strength, showing improvements in speed and cost, which is a highly practical advantage in the LLM era.
- The interpretable analysis on transferability and the relation of different activation features supports the motivation of the proposed method.

**Weaknesses:**

- The framework's first step requires a capabilities-specific in-domain dataset. For the tasks evaluated (MMLU, GSM, etc.), these datasets are readily available from established benchmarks. However, if one wishes to enhance a more abstract or novel capability, curating a high-quality, representative in-domain dataset becomes a significant, and potentially subjective, bottleneck.
- This paper should provide a few qualitative examples to illustrate the difference between data points in the "core set" (selected for most capabilities) and those that are task-specific (e.g., selected only when targeting CodeX)?
- The related work section correctly identifies Loss and Gradient-based Coreset Sampling Methods as a relevant category of data selection techniques. However, no gradient-based method was included as a baseline in the experimental comparisons.

**Questions:**

Same as the weaknesses.

---

> ### Author Response · Authors · 2025-11-25
> **Response to Reviewer 2hWE [Part 1/2]**
>
> Q1. The framework's first step requires a capabilities-specific in-domain dataset. For the tasks evaluated (MMLU, GSM, etc.), these datasets are readily available from established benchmarks. However, if one wishes to enhance a more abstract or novel capability, curating a high-quality, representative in-domain dataset becomes a significant, and potentially subjective, bottleneck.
>
> A1. We sincerely thank the reviewer for this constructive comment. In fact, to investigate the dependency of NAIT on the size of in-domain data, we have previously conducted relevant experiments in Section 5.1 (Page 8, Figure 3，Lines 343-363) of the original paper. These experiments employed an incremental testing approach with extremely small sample sizes (e.g., 16, 64, and 256 shots). The results demonstrated that even in the 16-shot scenario, models fine-tuned with data selected by NAIT generally outperformed random selection. Furthermore, when the sample size exceeded 64, the model performance was significantly superior to the random baseline. This provides strong evidence of NAIT's robustness even in scenarios with an extreme scarcity of in-domain data.
>
> To further address the reviewer's concern, we conducted additional experiments for the +NAIT (GSM) setting using three different random seeds for 16-shot sampling. The results are presented in the following table:
>
>
> **Table: Performance comparison of +NAIT (GSM) across different random seeds under the 16-shot setting.**
>
> | Baselines Benchmark→ | MMLU | MMLU-Pro | GSM | SVAMP |  H-Eval |  MBPP |  TydiQA |  XQuAD | BBH | AVG. | $\Delta$ |
> | :--- | :---: | :---: | :---: | :---: | :---: | :---: | :---: | :---: | :---: | :---: | :---: |
> | Alpaca-GPT4 | 46.87 | 21.89 | 14.63 | 39.00 | 27.87 | 51.58 | 39.48 | 42.99 | 39.94 | 36.03 | -
> | Random      | 47.14 | 21.43 | 14.13 | 35.67 | 25.55 | 47.35 | 44.16 | 46.56 | 39.21 | 35.69 | -0.94%
> | +NAIT (GSM)_seed 1  | 46.30 | 21.07 | 17.06 | 45.00 | 27.84 | 48.41 | 41.22 | 44.61 | 39.17 | 36.74 | +1.97%
> | +NAIT (GSM)_seed 2  | 46.81 | 21.89 | 17.13 | 43.00 | 29.27 | 47.88 | 44.13 | 46.85 | 37.59 | 37.17 | +3.16%
> | +NAIT (GSM)_seed 3  | 46.08 | 22.57 | 16.67 | 44.00 | 29.88 | 49.47 | 43.42 | 45.51 | 38.61 | 37.35 | +3.66%
>
> (1) Consistent Improvement in the Target Domain
>
> When GSM is used as the in-domain set, the model achieves significant and consistent performance gains on mathematical reasoning tasks (GSM, SVAMP) regardless of random seed variations. Results across all seeds substantially outperform both full fine-tuning and random selection baselines, with peak performances of 17.13 on GSM8K and 45.00 on SVAMP. This demonstrates that as long as the reference set aligns with the target domain, NAIT can stably identify and enhance domain-specific capabilities.
>
> (2) Robustness in Non-Target Domains
>
> As noted by the reviewer, we indeed observed certain performance fluctuations in non-target tasks (e.g., Multilingual Understanding on TydiQA, with a minimum score of 41.22). These fluctuations reflect the inherent variance characteristics associated with an extreme few-shot (16-shot) selection. However, it is crucial to emphasize that despite these local variations, the average performance (AVG) of NAIT remains robust across all seeds, showing an improvement of +1.97% to +3.66% over the baseline. This indicates that while NAIT enhances specific domain capabilities, it maintains an overall robustness superior to the random baseline.
>
>
>
> Q2. This paper should provide a few qualitative examples to illustrate the difference between data points in the "core set" (selected for most capabilities) and those that are task-specific (e.g., selected only when targeting CodeX)?
>
>
> A2. To address this concern, we have revised and expanded Section 5.3 (Selected IT Data Distribution) on Page 9. We have incorporated both qualitative and quantitative analyses, introducing specific case studies to visualize the data samples selected by NAIT. Furthermore, we provide comparisons with Embedding-based, Representation-based [1], and LESS [2] methods.
>
> The key updates are as follows:
>
> (1) Quantitative Distribution: As shown in Figure 7, NAIT identifies approximately 60% of the data as a shared "Core Set" across capabilities, while preserving high-complexity unique samples for specific tasks (e.g., GSM8K) as "Task-Specific Data."
>
> (2) Qualitative Examples: As presented in the newly added Table 12, we provide specific qualitative samples to elucidate the distinctions. In contrast to embedding-based methods, which tend to select data based on surface textual similarity, NAIT prioritizes samples requiring multi-step logical reasoning and programmatic synthesis (e.g., algorithm design problems) for the core set. This demonstrates that our method aligns more effectively with the deep semantic requirements of the target capabilities, rather than merely matching surface patterns.

---

> ### Author Response · Authors · 2025-11-25
> **Response to Reviewer 2hWE [Part 2/2]**
>
> Q3. The related work section correctly identifies Loss and Gradient-based Coreset Sampling Methods as a relevant category of data selection techniques. However, no gradient-based method was included as a baseline in the experimental comparisons.
>
> A3. We fully recognize the significance of gradient-based subset selection methods, such as LESS [2], in the field of sample importance estimation. However, we would like to first clarify why we initially did not consider such methods: such methods typically necessitate additional backpropagation steps to compute gradient signals, resulting in significantly higher computational costs compared to schemes based solely on forward features. Given that this paper focuses primarily on efficient data selection within resource-constrained scenarios, we initially prioritized baselines based on forward activations, as they offer comparability while avoiding the overhead of gradient computation.
>
> Nevertheless, we find the reviewer’s insight highly valuable. Consequently, to provide a comprehensive evaluation, we have incorporated comparisons with the gradient-based method LESS [2], as well as Embedding-based and Representation-based [1] approaches. The updated results are presented in the table below:
>
> (1) Performance & Generalization
>
> Experimental results indicate that while LESS achieves high scores on specific target tasks (e.g., MMLU), it exhibits a tendency to overfit the target validation set, resulting in suboptimal generalization on non-target tasks. In contrast, NAIT demonstrates superior cross-task transferability while maintaining competitive performance on target tasks. Consequently, NAIT outperforms LESS in terms of overall average performance (e.g., when targeting MMLU, the average score for LESS is 39.94, whereas NAIT achieves 41.31).
>
> (2) Computational Efficiency
>
> We have added a detailed cost comparison in Table 5 (also can see below). The results demonstrate that NAIT successfully circumvents the high overhead associated with gradient computation (requiring no backpropagation). It achieves the best overall average performance while significantly reducing computational costs. This further validates NAIT's ability to strike a superior balance between selection efficiency and the capture of deep model capabilities in resource-constrained scenarios.
>
>
> **Table: Performance comparison of baselines of activating targeted ability using 10% of the IT data. The detail experiment setting shown on the Appendix E, Page 17**
>
> | Method | MMLU | MMLU-Pro | TydiQA | XQuAD | BBH | AVG |
> | :--- | :---: | :---: | :---: | :---: | :---: | :---: |
> | *+MMLU* | | | | | | |
> | Embedding | 46.29 | 21.07 | 42.97 | 44.73 | 36.67 | 38.35 |
> | Representation-based | 45.57 | 23.0 | 46.78 | 46.85 | 38.15 | 40.07 |
> | LESS | **48.12** | **24.50** | 43.42 | 45.51 | 38.15 | 39.94 |
> | Nait | 47.81 | 23.61 | **47.16** | **49.47** | **38.52** | **41.31** |
> | *+TydiQA* | | | | | | |
> | Embedding-based | **47.49** | 22.54 | 42.4 | 46.53 | 38.43 | 39.48 |
> | Representation-based | 45.91 | 21.04 | 40.02 | 46.0 | 36.02 | 37.80 |
> | LESS | 44.68 | 22.21 | 45.24 | 47.67 | 37.69 | 39.50 |
> | Nait | 46.17 | **22.82** | **47.78** | **49.23** | **40.00** | **41.20** |
> | *+BBH* | | | | | | |
> | Embedding | 47.51 | 22.86 | 43.25 | 46.66 | 36.76 | 39.41 |
> | Representation-based | 45.81 | 21.11 | 36.78 | 44.61 | 37.59 | 37.18 |
> | LESS | 46.85 | **23.75** | **46.56** | 47.67 | 39.72 | 39.61 |
> | Nait | **47.78** | 23.36 | 45.93 | **48.46** | **40.46** | **41.20** |
>
>
> **Table: PEfficiency Comparison of Different Methods on NVIDIA A800 80GB with Batch Size set to 8**
>
> > **Note:** API costs follow the official OpenAI pricing ($2.00/million input tokens, $8/million output tokens for GPT-4.1); GPU costs are estimated based on the Google Cloud pricing ($1.15 per GPU hour for NVIDIA A800 80GB).
>
> | Method | Externally-Independent | Time | Cost |
> | :--- | :---: | :---: | :---: |
> | AlpaGasus | ✗ | 19.07h | $178.02 |
> | Q2Q | ✗ | 3.52h | $4.05 |
> | LESS | ✓ | 9.86h | $11.33 |
> | SelectIT | ✓ | 23.20h | $26.68 |
> | **NAIT** | ✓ | **1.32h** | **$1.52** |
>
>
> Reference:
>
> [1] Evaluation of similarity-based explanations. In 9th International Conference on Learning Representation. ICLR, 2021.
>
> [2] LESS: selecting influential data for targeted instruction tuning. ICML, 2024.

---

### Author Response · Authors · 2025-11-25
**General Response to Reviewers**

We thank all reviewers for their constructive comments. Please note that the newly added content in the revised manuscript is highlighted in blue.

---

### Author Response · Authors · 2025-12-04
**Summary of Rebuttal Updates**

Dear Area Chair and Reviewers,

We acknowledge the recent communication from the Program Chairs regarding the review process adjustments. We fully understand the complexity of the situation and support the difficult decisions made to maintain the integrity and fairness of the ICLR community.

Given that the review scores have been reverted, we would like to take this opportunity to provide a concise summary of the **significant updates and additional experiments** conducted during the discussion period. We hope this summary assists the AC in assessing the current state of our paper, which we believe has been substantially strengthened in response to the reviewers' constructive feedback.

**1. Inclusion of Strong Baselines (Addressing R2hWE, Rwy8Y, RSZW9)**
*   **Gradient-based Comparison:** We implemented and compared NAIT against **LESS (ICML 2024)**. Results show NAIT achieves superior average performance while being significantly more efficient (requiring **no backpropagation** and reducing computational time by approx. **85%** compared to LESS).
*   **Other Baselines:** We also added Embedding-based and Representation-based baselines, demonstrating NAIT's advantage in capturing deep capability features over surface-level similarity.

**2. Validation on SOTA Models & Hard Benchmarks (Addressing R76SK, RSZW9)**
*   **Modern Base Models:** We extended our evaluation to **Qwen-2.5-7B**. NAIT consistently outperformed random selection (+3.83%) and full fine-tuning, proving its effectiveness on stronger, state-of-the-art base models.
*   **Logic-Intensive Benchmarks:** To address concerns about OOD generalization, we evaluated on **LiveCodeBench, MATH-500, Minerva-Math, MBPP, and MMLU-Pro**. NAIT demonstrated strong generalization, particularly in coding and mathematical reasoning tasks.

**3. Empirical Validation of Core Hypothesis (Addressing R76SK)**
*   We conducted a "High vs. Low Activation" experiment. Results showed that data with high neuron activation scores improved performance (+3.35%), while low-scoring data caused performance collapse (-17.54%). This empirically substantiates our core hypothesis that activation patterns are causal to data effectiveness.

**4. Methodological Clarifications & Robustness**
*   **Data Union Strategy:** We clarified our approach for multi-capability scenarios, showing that a union of selected subsets outperforms feature averaging.
*   **Robustness:** We verified NAIT's stability under extreme few-shot settings (16-shot) with multiple random seeds.
*   **Algorithm Details:** A detailed Algorithm Box and cost analysis table have been added to the revised appendix.

We are confident that these revisions address the primary concerns regarding baselines, model obsolescence, and theoretical grounding. We thank the AC for their time and extra effort in evaluating our work under these unique circumstances.

Sincerely,

The Authors

---

### Meta-Review · Area_Chair_8JPC · 2026-01-07

**Summary:**

The paper proposes NAIT, a framework for selecting high-quality instruction tuning (IT) data for large language models by leveraging neuron activation patterns. The method extracts activation features from small in-domain datasets, compresses them via PCA, and scores candidate samples based on alignment with these features. Experiments across multiple benchmarks demonstrate that NAIT can achieve comparable or superior performance using only a fraction of the training data, with notable gains in cost-efficiency and interpretability.

**Reviewer Concerns:**

- Dependence on In-Domain Reference Sets: NAIT requires representative datasets for each target capability. While benchmarks like GSM or MMLU provide such sets, curating data for novel or abstract capabilities may be challenging and subjective.
- Limited Baselines: The experimental comparisons omit several strong recent methods (e.g., LESS, gradient-based approaches, embedding similarity baselines), which weakens claims of superiority.
- Design Choices Not Fully Justified: Important details such as which layers’ activations are used, how alignment scores are computed, and how multi-capability features are combined are insufficiently explained. This reduces reproducibility and leaves open questions about scalability.
- Benchmark Coverage: The evaluation relies on a relatively narrow set of benchmarks. More comprehensive suites (e.g., MMLU-Pro, GPQA, HumanEval, Minerva-Math) would strengthen claims of generalizability.
- Model Selection: Experiments are conducted primarily on LLaMA and Mistral, which some reviewers note are weaker than other contemporary open-source models (e.g., Qwen series). This choice may limit persuasiveness of results.

**Reviewer Scores:**

Reviewer 2hWE may have a chance to raise the score.

---

### Decision · Program_Chairs · 2026-01-26

Accept (Poster)